# OPTIMA: OPTIMAL ONE-SHOT PRUNING FOR LLMS VIA QUADRATIC PROGRAMMING RECONSTRUCTION

## ABSTRACT

Post-training model pruning is a promising solution, yet it faces a trade-off: *simple heuristics that zero weights are fast but degrade accuracy, while principled joint optimization methods recover accuracy but are computationally infeasible at modern scale*. One-shot methods such as SparseGPT offer a practical trade-off in optimality by applying efficient, approximate heuristic weight updates. To close this gap, we introduce OPTIMA, a practical one-shot post-training pruning method that balances accuracy and scalability. OPTIMA casts layer-wise weight reconstruction after mask selection as independent, column-wise Quadratic Programs (QPs) that share a common layer Hessian. Solving these QPs yields the per-column globally optimal update with respect to the reconstruction objective given the estimated Hessian. The shared-Hessian structure makes the problem highly amenable to batching on accelerators. We implement an accelerator-friendly QP solver that accumulates one Hessian per layer and solves many small QPs in parallel, enabling one-shot post-training pruning at scale on a single accelerator without fine-tuning. OPTIMA integrates with existing mask selectors and consistently improves zero-shot performance across multiple LLM families and sparsity regimes, yielding up to 2.53% absolute accuracy improvement. On an NVIDIA H100, OPTIMA prunes a 8B-parameter transformer end-to-end in 40 hours with 60 GB peak memory. Together, these results set a new state-of-the-art accuracy-efficiency trade-offs for one-shot post-training pruning.[1]

## 1 INTRODUCTION

Large language models (LLMs) deliver unprecedented capabilities across a wide array of natural language tasks (Team et al., 2024a; Comanici et al., 2025; Touvron et al., 2023; Guo et al., 2025). However, their rapidly growing parameter counts create severe compute and memory burdens that complicate deployment and inference. Post-training one-shot pruning (Hoefler et al., 2021), which removes parameters from a pretrained model with only a small calibration dataset, promises to reduce these costs, yet it faces a fundamental trade-off: very fast, heuristic schemes that simply zero weights (e.g., Wanda (Sun et al., 2023) and ProxSparse (Liu et al., 2025)) are cheap but often incur noticeable accuracy losses, while principled second-order approaches (e.g., Optimal Brain Surgeon (Hassibi et al., 1993)) recover accuracy but are computationally infeasible at modern LLM scales. One-shot approximations such as SparseGPT (Frantar & Alistarh, 2023) and related heuristics (Ilin & Richtarik, 2025) try to navigate this middle ground, but they sacrifice reconstruction optimality and therefore leave headroom in accuracy.[2]

In this paper we introduce OPTIMA, a practical one-shot post-training pruning framework that closes much of this gap by combining principled optimality with accelerator-grade efficiency. The core idea is a precise reformulation of the layer-wise reconstruction step that follows mask selection. That is, after fixing a binary mask for a weight matrix, the reconstruction (least-squares) objective decomposes across rows and each row's update can be written as a small quadratic program (QP). Crucially, every row in the same layer shares the same Hessian matrix $H = X^\top X$, while the linear constraints differ only according to which entries the mask removes. This shared-Hessian, row-wise QP structure yields two immediate benefits: (1) per-row global optimality for the reconstruction objective (given the estimated Hessian), and (2) uniform problem structure that enables massive batching and parallelism on off-the-shelf ML accelerators (GPUs/TPUs).

---

[1]The code and data for OPTIMA is available at https://anonymous.4open.science/r/OPTIMA-ICLR2026

[2]For a more detailed discussion of the related work, see Appendix B

Figure 1: `OPTIMA` generates a shared Hessian among the different rows of the pruned weight using a small calibration dataset. Then, the weights in different rows will be updated in parallel using a QP solver and the shared Hessian.

Realizing this formulation in practice requires careful numerical and systems engineering. We adopt a first-order primal–dual QP solver (rAPDHG (Lu & Yang, 2023)) that is well-suited to our constrained problems and whose critical operations reduce to matrix–vector products with the shared Hessian. This makes the inner loops extremely efficient on accelerators. We further avoid explicit dense equality matrices by enforcing fixed entries via tight bounds, accumulate layer Hessians incrementally from calibration sequences to save memory, and solve rows in batches so thousands of small QPs are processed in parallel. These implementation choices make `OPTIMA` not only theoretically principled but also practical to run on a single accelerator.

We evaluate `OPTIMA` across multiple model families (LLaMA, Gemma, and others) and sparsity regimes (unstructured and 2:4 semi-structured sparsity). `OPTIMA` is modular and plugs into existing mask selectors (e.g., Wanda, SparseGPT, Thanos), consistently improving zero-shot performance. Across eight zero-shot downstream benchmarks in Language Model Evaluation Harness, we observe up to 2.53 percentage-point absolute gains on downstream tasks without any post-pruning fine-tuning. In summary, our contributions are:

- We present a row-wise QP reformulation of the post-training reconstruction problem that yields per-row global optimality under a shared-Hessian model and is provably equivalent to the least-squares objective after mask selection (section 3).
- We design and implement an accelerator-friendly QP solver pipeline that accumulates a single Hessian per layer, enforces mask constraints via bounds, batches thousands of row QPs, and leverages rAPDHG/MPAX for efficient execution on GPUs/TPUs (detailed in Algorithm 1).
- We show the modularity of `OPTIMA`, which can be used as a drop-in weight-update step with common mask selection algorithms (Wanda, SparseGPT, Thanos), consistently improving their accuracy without fine-tuning (section 4).
- We provide extensive empirical evidence and practical measurements. `OPTIMA` yields substantial average accuracy gains across tasks and model sizes (up to 2.53%), demonstrates robustness at high sparsity (up to 60%), and can prune billion-parameter models on a single H100 in less than 40 hours.

## 2 PRELIMINARIES

Post-training pruning (PTP) compresses pre-trained models without retraining, using a small calibration dataset to produce a sparse model that preserves performance. To make PTP tractable, the problem is decomposed into independent layer-wise subproblems. For layer $l$, the goal is to find a binary sparsity mask $\mathbf{M}_l$ and updated weights $\hat{\mathbf{W}}_l$ that minimize the output reconstruction error given original weights $\mathbf{W}_l$ and input activations $\mathbf{X}_l$. This task can be formulated as in Equation 1, where $\odot$ denotes the Hadamard product, and $\mathbf{M}_l$ is a binary tensor of the same shape as $\mathbf{W}_l$ with 0s for pruned weights and 1s for retained ones. Equation 1 is solved sequentially across layers, with $\mathbf{X}_l$ as the pruned output from layer $l-1$. Finding the optimal $\mathbf{M}_l$ is NP-hard, motivating heuristics.

$$\underset{\mathbf{M}_l, \hat{\mathbf{W}}_l}{\operatorname{argmin}} \|\mathbf{X}_l \mathbf{W}_l - \mathbf{X}_l (\mathbf{M}_l \odot \hat{\mathbf{W}}_l)\|_F^2 \qquad (1)$$

A common heuristic decouples mask selection from weight updates. After selecting $\mathbf{M}_l$ (e.g., by magnitude), the problem simplifies to Equation 2, which is a convex least-squares problem, but solving it directly is computationally expensive for large LLM weights.

$$\min_{\hat{\mathbf{W}}_l} \|\mathbf{X}_l\mathbf{W}_l - \mathbf{X}_l(\mathbf{M}_l \odot \hat{\mathbf{W}}_l)\|_F^2 \tag{2}$$

Consequently, many methods employ strategies to circumvent the expensive weight update step. For example, **Wanda** Sun et al. (2023) avoids weight updates altogether, simply setting the selected weights to zero. However, other methods such as **SparseGPT** Frantar & Alistarh (2023) and **Thanos** (Ilin & Richtarik, 2025) adopt a compromise, performing a more complex update but only on a small subset of the weights. These heuristics trade off optimality for computational feasibility.

## 3 OPTIMA: OPTIMAL WEIGHT UPDATES VIA QUADRATIC PROGRAMMING

To overcome the challenges of weight update in LLM pruning, we propose OPTIMA, a novel approach that enables the efficient and optimal update of **all** remaining weights once the pruning mask $\mathbf{M}_l$ has been chosen.

We achieve this by reformulating the least-squares problem as a set of independent Quadratic Programs (QPs) that can be solved in parallel on hardware accelerators like GPUs or TPUs using iterative methods. Specifically, we derive both a linearly constrained QP formulation and an equivalent unconstrained formulation. While the unconstrained form can be useful for optimizers restricted to such problems or in cases where it can be solved more efficiently, our implementation focuses on the constrained QP formulation, which is more amenable to GPU/TPU acceleration.

### 3.1 REFORMULATION AS A QUADRATIC PROGRAM WITH LINEAR CONSTRAINTS

As discussed in section 2, our goal is to minimize the problem defined in Equation 2. The Frobenius norm objective function in Equation 2 is separable by the columns of the weight matrix.[3] We can therefore solve the optimization problem for each column independently.

Let $\mathbf{w}_j$ be the $j$-th column of the original weight matrix $\mathbf{W}_l$, and let $\hat{\mathbf{w}}_j$ be the corresponding column in the updated matrix $\hat{\mathbf{W}}_l$. The mask for this column is $\mathbf{m}_j$. The optimization for this single column can be formulated as in Equation 3.

$$\min_{\hat{\mathbf{w}}_j} \|\mathbf{X}_l\mathbf{w}_j - \mathbf{X}_l(\mathbf{m}_j \odot \hat{\mathbf{w}}_j)\|_2^2 \tag{3}$$

By defining the change in the weight column as $\Delta\mathbf{w}_j = (\mathbf{m}_j \odot \hat{\mathbf{w}}_j) - \mathbf{w}_j$, the objective can then be rewritten in terms of this change as in Equation 4 in standard quadratic form.

$$\min_{\Delta\mathbf{w}_j} \|-\mathbf{X}_l\Delta\mathbf{w}_j\|_2^2 = \min_{\Delta\mathbf{w}_j} \Delta\mathbf{w}_j^T(\mathbf{X}_l^T\mathbf{X}_l)\Delta\mathbf{w}_j \tag{4}$$

The constraints on $\Delta\mathbf{w}_j$ in Equation 4 are determined by the mask $\mathbf{m}_j$. Let $\mathcal{S}_j$ be the set of indices where the mask is zero (i.e., weights to be pruned). For each index $i \in \mathcal{S}_j$, the corresponding entry in the updated weight vector, $(\hat{\mathbf{w}}_j)_i$, must be zero. This imposes a linear constraint on the change vector, as shown in Equation 5.

$$(\mathbf{m}_j \odot \hat{\mathbf{w}}_j)_i = 0 \implies (\Delta\mathbf{w}_j)_i = -(\mathbf{w}_j)_i \quad \forall i \in \mathcal{S}_j \tag{5}$$

The entries of $\Delta\mathbf{w}_j$ for the unpruned weights (where $m_{ij} = 1$) remain as free variables to be optimized.

For each column $j$ of the weight matrix, we have a QP of the form represented in Equation 6, where $\mathbf{H} = \mathbf{X}_l^T\mathbf{X}_l$ is the Hessian matrix, which is positive semi-definite and shared across all column-wise

---

[3]Once the mask has been chosen, the weight reconstruction is separable for each column

problems. The fact that the Hessian is shared among all columns, and only the constraints change, makes it very easy to parallelize on accelerators such as GPUs and TPUs.

$$\begin{aligned} \underset{\Delta\mathbf{w}_j}{\text{minimize}} \quad & \Delta\mathbf{w}_j^T\mathbf{H}\Delta\mathbf{w}_j \\ \text{subject to} \quad & (\Delta\mathbf{w}_j)_i = -(\mathbf{w}_j)_i, \forall i \in \mathcal{S}_j \end{aligned} \tag{6}$$

## 3.2 REFORMULATION AS AN UNCONSTRAINED QUADRATIC PROGRAM

As an alternative to the constrained formulation in Equation 6, we can reformulate each column-wise problem as an unconstrained quadratic program. This can be useful in settings where solvers are optimized for unconstrained problems or when eliminating constraints enables more efficient optimization. Although our implementation adopts the constrained approach for reasons discussed below, we include the unconstrained version for completeness.

The key idea is to eliminate the equality constraints in Equation 5 by substituting them directly into the objective. For a given $j$, define $\mathcal{I}_j$ as the set of indices where the mask is one (i.e., unpruned weights), and let $\mathcal{S}_j$ denote the complement set (i.e., pruned weights, where the mask is zero).

We reorder the entries of the change vector $\Delta\mathbf{w}_j$ and the shared Hessian matrix $\mathbf{H} = \mathbf{X}_l^T\mathbf{X}_l$ based on this partitioning, as shown in Equation 7.

$$\Delta\mathbf{w}_j = \begin{bmatrix} \Delta\mathbf{w}_{\mathcal{I}_j} \\ \Delta\mathbf{w}_{\mathcal{S}_j} \end{bmatrix}, \quad \mathbf{H} = \begin{bmatrix} \mathbf{H}_{\mathcal{I}_j\mathcal{I}_j} & \mathbf{H}_{\mathcal{I}_j\mathcal{S}_j} \\ \mathbf{H}_{\mathcal{S}_j\mathcal{I}_j} & \mathbf{H}_{\mathcal{S}_j\mathcal{S}_j} \end{bmatrix} \tag{7}$$

As established in Equation 5, the entries of $\Delta\mathbf{w}_j$ corresponding to $\mathcal{S}_j$ are fixed: $(\Delta\mathbf{w}_j)_i = -(\mathbf{w}_j)_i$ for all $i \in \mathcal{S}_j$. Substituting these fixed values into the quadratic objective yields the expanded form in Equation 8.

$$\Delta\mathbf{w}_j^T\mathbf{H}\Delta\mathbf{w}_j = \Delta\mathbf{w}_{\mathcal{I}_j}^T\mathbf{H}_{\mathcal{I}_j\mathcal{I}_j}\Delta\mathbf{w}_{\mathcal{I}_j} + 2\Delta\mathbf{w}_{\mathcal{I}_j}^T\mathbf{H}_{\mathcal{I}_j\mathcal{S}_j}\Delta\mathbf{w}_{\mathcal{S}_j} + \Delta\mathbf{w}_{\mathcal{S}_j}^T\mathbf{H}_{\mathcal{S}_j\mathcal{S}_j}\Delta\mathbf{w}_{\mathcal{S}_j} \tag{8}$$

Since $\Delta\mathbf{w}_{\mathcal{S}_j} = -\mathbf{w}_{\mathcal{S}_j}$, we substitute this to obtain the unconstrained objective in Equation 9.

$$\min_{\Delta\mathbf{w}_{\mathcal{I}_j}} \left( \Delta\mathbf{w}_{\mathcal{I}_j}^T\mathbf{H}_{\mathcal{I}_j\mathcal{I}_j}\Delta\mathbf{w}_{\mathcal{I}_j} - 2\Delta\mathbf{w}_{\mathcal{I}_j}^T\mathbf{H}_{\mathcal{I}_j\mathcal{S}_j}\mathbf{w}_{\mathcal{S}_j} + \mathbf{w}_{\mathcal{S}_j}^T\mathbf{H}_{\mathcal{S}_j\mathcal{S}_j}\mathbf{w}_{\mathcal{S}_j} \right) \tag{9}$$

The final term in Equation 9 is constant with respect to the optimization variable $\Delta\mathbf{w}_{\mathcal{I}_j}$ and can therefore be omitted. This results in the unconstrained quadratic program in Equation 10.

$$\underset{\Delta\mathbf{w}_{\mathcal{I}_j}}{\text{minimize}} \quad \Delta\mathbf{w}_{\mathcal{I}_j}^T\mathbf{Q}_j\Delta\mathbf{w}_{\mathcal{I}_j} + \mathbf{c}_j^T\Delta\mathbf{w}_{\mathcal{I}_j} \tag{10}$$

where the problem-specific matrix and vector are defined as:

$$\mathbf{Q}_j = \mathbf{H}_{\mathcal{I}_j\mathcal{I}_j}, \quad \mathbf{c}_j = -2\mathbf{H}_{\mathcal{I}_j\mathcal{S}_j}\mathbf{w}_{\mathcal{S}_j} \tag{11}$$

This formulation eliminates the need for explicit constraints, but introduces column-dependent variation in problem dimensions. Specifically, the size of $\mathbf{Q}_j$ and $\mathbf{c}_j$ varies with the number of unpruned weights in each column. Consequently, the unconstrained QPs have heterogeneous shapes and objectives across columns, making them more difficult to batch and parallelize efficiently on accelerators like GPUs or TPUs. This motivates our choice to adopt the constrained formulation in Equation 6, where the problem structure is uniform and well-suited for high-throughput parallel execution.

## 3.3 SOLVING THE QUADRATIC PROGRAMS

With the constrained QP formulation established, we now select a solver, whose efficiency is crucial for runtime and scalability on parallel hardware like GPUs and TPUs. Our QP, with its shared Hessian $\mathbf{H}$ and simple bounds, suits specialized modern solvers. We adopt the state-of-the-art Restarted Accelerated Primal-Dual Hybrid Gradient (rAPDHG) algorithm (Lu & Yang, 2023), a first-order method effective

---

**Algorithm 1 Layer-wise Pruning with Batched Row-wise Quadratic Programming**

---

**Input:** Pre-trained LLM $\mathcal{M}$, calibration data $\mathbf{X}$, pruning masks $\mathcal{M}_{\text{ask}}$, QP solver $\mathcal{S}$, batch size $B$.
**Output:** Pruned and updated LLM $\hat{\mathcal{M}}$, updated masks $\hat{\mathcal{M}}_{\text{ask}}$.

1  **for each layer** $L$ in the LLM $\mathcal{M}$ **do**
2    Initialize Hessian estimate $\mathbf{H} \leftarrow 0$.                    ▷ Initialize covariance matrix
3    **for each calibration sample** $x \in \mathbf{X}$ **do**
4      $y \leftarrow L(x)$                           ▷ Forward pass for one sequence
5      $\mathbf{H} \leftarrow \mathbf{H} + y^T y$                       ▷ Accumulate covariance
6    **end for**
7    Store intermediate inputs $\{\mathbf{X_W} \mid \mathbf{W} \in L\}$ from a forward pass of $L(\mathbf{X})$.
8    **for each weight matrix** $\mathbf{W}$ in layer $L$ **do**
9      Retrieve corresponding mask $\mathbf{M} \in \mathcal{M}_{\text{ask}}$.
10     Partition the rows of $\mathbf{W}$ into batches of size $B$.
11     **for each batch of rows** $\{\mathbf{w}_j\}_{j=1}^{B}$ **in parallel do**
12       **for each row** $\mathbf{w}_j$ in the batch **do**
13         $\mathcal{S}_j \leftarrow \{i \mid \mathbf{M}_{j,i} = 0\}$                   ▷ Indices of pruned entries
14         Define QP:

$$\min_{\Delta \mathbf{w}_j} \Delta \mathbf{w}_j^T \mathbf{H} \Delta \mathbf{w}_j \tag{12}$$

$$\text{s.t. } (\Delta \mathbf{w}_j)_i = -(\mathbf{w}_j)_i, \forall i \in \mathcal{S}_j$$

15       **end for**
16       $\{\Delta \mathbf{w}_j\}_{j=1}^{B} \leftarrow \mathcal{S}(\mathbf{H}, \{\mathbf{w}_j\}_{j=1}^{B}, \{\mathcal{S}_j\}_{j=1}^{B})$
17       Update weights: $\mathbf{w}_j \leftarrow \mathbf{w}_j + \Delta \mathbf{w}_j, \quad \forall j$
18     **end for**
19   **end for**
20   $\mathbf{X} \leftarrow L(\mathbf{X})$                         ▷ Update activations for next layer
21 **end for**

**Return:** Updated model $\hat{\mathcal{M}}$, updated masks $\hat{\mathcal{M}}_{\text{ask}}$.

---

here for three reasons: (1) its bottleneck—matrix-vector multiplications with $\mathbf{H}$ and its transpose—runs efficiently on GPUs/TPUs; (2) it achieves provably optimal linear convergence; and (3) a high-performance, open-source JAX-based implementation is available in MPAX (Lu et al., 2024), designed for GPU/TPU execution. This enables parallel solving of thousands of column-wise QPs, leveraging the shared structure.

## 3.4 EFFICIENT IMPLEMENTATION

Naively implementing the optimization problem in Equation 6 is computationally expensive and incurs substantial memory overhead. These costs, however, can be greatly reduced through a series of optimization techniques. In the following, we describe the strategies we employ to solve the QPs efficiently on a single GPU, even for very large LLMs. Additionally, a detailed algorithm of our implementation is provided in Algorithm 1.

**Equality constraints.** Directly encoding the constraints from Equation 5 into the standard quadratic objective leads to a prohibitively large matrix of equalities, even though these constraints merely fix individual variables to constant values. To avoid constructing such large matrices, we instead enforce the constraints by setting upper and lower bounds on the corresponding variables. In particular, fixing the bounds of $(\Delta w_j)_i$ to $-(w_j)_i$ effectively locks the variable to the desired value, without incurring the overhead of explicit equality matrices.

**Batching QP problems.** In memory-limited scenarios, the optimization problems for all columns of the weight matrices may not fit on a single GPU. To address this, we employ a batching strategy that solves a subset of QP problems at a time. This approach reduces memory overhead while still leveraging the efficiency of solving multiple QPs in parallel. As a result, our method enables pruning of large LLMs even on a single GPU.

**Hessian calculation.** For each layer, the Hessian matrix can be estimated as the covariance of the dense model's inputs across multiple sequences. Suppose the output tensor is $Y \in \mathbb{R}^{b \times s \times d}$, where $b$ is the number of sequences, $s$ is the sequence length, and $d$ is the output dimension of the layer. To compute the covariance directly, we would first reshape $Y$ into $\hat{Y} \in \mathbb{R}^{bs \times d}$, effectively stacking all tokens from all sequences into a single matrix, and then evaluate $\hat{Y}^T \hat{Y}$.

While this formulation is straightforward, it requires storing the full $Y$ in accelerator memory, which becomes prohibitively expensive for large $b$ and $s$, often causing out-of-memory errors. To make the computation feasible, we observe that the covariance can be accumulated incrementally. Specifically,

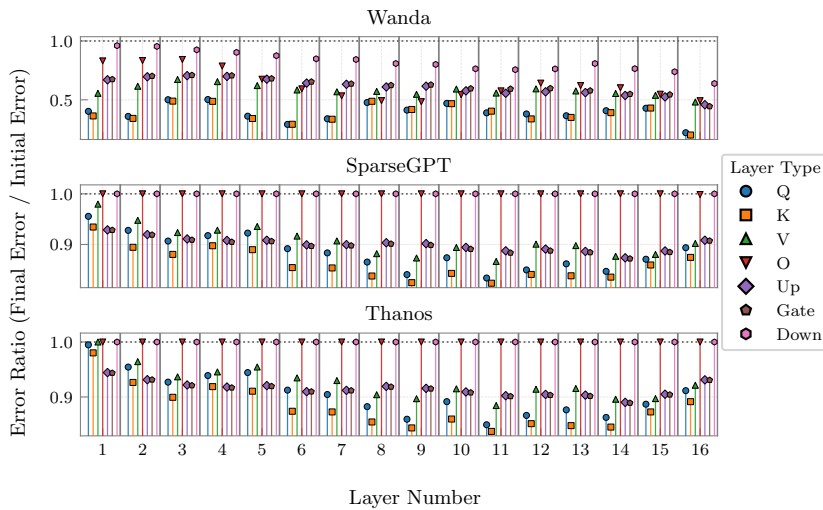

Figure 2: Relative error reduction on OPTIMA in comparison to Wanda, SparseGPT, and Thanos for LLaMA-3.2 1B.

$Y$ can be decomposed into $b$ smaller matrices, $y_i \in \mathbb{R}^{s \times d}$, each corresponding to the output of a single sequence. Instead of materializing $\hat{Y}$, we compute $y_i^T y_i$ for each sequence separately and sum the results as in $H \approx \sum_{i=1}^{b} y_i^T y_i$. This decomposition yields the same result as computing $\hat{Y}^T \hat{Y}$ directly, but avoids the need to store the entire $Y$ at once, making the approach scalable to very large LLMs.

## 4 EXPERIMENTS

**Model, datasets, and evaluation.** We evaluate OPTIMA on LLaMA 3.1, LLaMA 3.2 (Dubey et al., 2024), Gemma 2 (Team et al., 2024b), and Gemma 3 (Team et al., 2025) family of models. Model accuracy is assessed on a range of zero-shot downstream tasks, including MMLU (Hendrycks et al., 2020), Piqa (Bisk et al., 2020), Arc-Easy, Arc-Challenge (Clark et al., 2018), WinoGrande (Sakaguchi et al., 2021), and OpenBookQA (Mihaylov et al., 2018), all of which are commonly used to evaluate LLM compression (Mozaffari et al., 2025a; Sun et al., 2023). For zero-shot evaluations, we utilize the Language Model Evaluation Harness (Gao et al., 2024) framework. In line with prior work (Sun et al., 2023; Frantar & Alistarh, 2023; Mozaffari et al., 2025a), we also report the perplexity of the models on a language modeling task on the WikiText2 (Merity et al., 2016) dataset.

**Baselines.** We compare OPTIMA against state-of-the-art one-shot pruning methods, including Wanda (Sun et al., 2023), SparseGPT (Frantar & Alistarh, 2023), Thanos (Ilin & Richtarik, 2025), and ProxSparse (Liu et al., 2025) and show how OPTIMA can improve the performance of all these pruning methods across different models and datasets. Additional details about the hyperparameters used in OPTIMA is provided in Appendix C, and the sensitivity of OPTIMA to the calibration dataset size can be found in Appendix D. In terms of memory reductions and speedup, our method is guaranteed to achieve the same performance as other pruning methods such as Wanda and SparseGPT, since the sparsity pattern in these methods stays intact.

**Model quality.** We evaluate the accuracy of OPTIMA and other state-of-the-art pruning methods across 2:4 and unstructured sparsity benchmarks. Wanda is a mask selection algorithm, that does not provide any weight update mechanism for the weights. SparseGPT and Thanos, on the other hand, update the weight values in addition to searching for the best mask. We couple OPTIMA weight update with the masks generated using each of these methods and compare the resulting performance of the models.

Table 1 summarizes the the performance metrics for Wanda, SparseGPT, and Thanos with and without the OPTIMA update mechanism for 50% unstructured sparsity. It can be seen that models pruned with OPTIMA weight update scheme consistently outperform the methods using weight update methods, providing up to 1.80% average accuracy improvement across six downstream tasks (Gemma-3-1B).

| Model | Mask Selection | Weight Update | Perplexity | Metrics (%) | | | | | | |
|---|---|---|---|---|---|---|---|---|---|---|
| | | | | MMLU | PIQA | Arc-E | Arc-C | Wino | OpenQA | **Average** |
| LLaMA 3.1 8B | Dense | - | 5.84 | 63.57 | 80.09 | 81.44 | 51.37 | 73.48 | 33.40 | 63.89 |
| | Wanda | – | 9.64 | 47.79 | 75.68 | 72.56 | 40.70 | 70.09 | 27.40 | 55.70 |
| | Wanda | OPTIMA | 9.37 | **48.85** | **76.71** | **73.82** | **42.32** | **70.32** | **28.20** | **56.70** |
| | SparseGPT | SparseGPT | 9.30 | **51.32** | 76.19 | 73.02 | 41.27 | **70.88** | **29.40** | 57.01 |
| | SparseGPT | OPTIMA | 9.33 | 49.31 | **76.61** | **74.28** | **42.83** | 70.88 | 28.20 | **57.02** |
| | Thanos | Thanos | 9.27 | **50.36** | **77.04** | **74.92** | **42.58** | **70.96** | **30.00** | **57.64** |
| | Thanos | OPTIMA | 9.35 | 50.17 | 76.50 | 74.16 | 41.89 | 70.24 | 28.40 | 56.89 |
| LLaMA 3.2 1B | Dense | – | 9.75 | 36.92 | 74.27 | 65.53 | 31.31 | 60.30 | 26.20 | 49.09 |
| | Wanda | – | 23.51 | 26.35 | 65.18 | 52.10 | 23.81 | 54.62 | 18.00 | 40.01 |
| | Wanda | OPTIMA | 18.84 | **27.69** | **67.08** | **52.61** | **24.74** | **55.64** | **20.20** | **41.33** |
| | SparseGPT | SparseGPT | 18.84 | 25.71 | 67.85 | 54.29 | **26.54** | **57.70** | 22.00 | 42.35 |
| | SparseGPT | OPTIMA | 18.09 | **26.95** | **68.01** | **54.59** | 25.85 | 56.91 | **24.00** | **42.72** |
| | Thanos | Thanos | 19.70 | 25.37 | 67.63 | 52.99 | **27.13** | 54.38 | 22.20 | 41.62 |
| | Thanos | OPTIMA | 18.77 | **25.99** | **68.23** | **53.49** | 26.45 | **55.88** | 21.60 | **41.94** |
| LLaMA 3.2 3B | Dense | – | 7.81 | 54.13 | 76.55 | 74.28 | 42.75 | 69.38 | 30.60 | 57.95 |
| | Wanda | – | 12.92 | 40.79 | 72.03 | 65.45 | 32.34 | 63.69 | 25.40 | 49.95 |
| | Wanda | OPTIMA | 12.24 | **43.11** | **72.47** | **66.50** | **33.53** | **66.38** | **26.20** | **51.37** |
| | SparseGPT | SparseGPT | 12.32 | 37.96 | **73.45** | 65.19 | 33.02 | 66.38 | 25.20 | 50.20 |
| | SparseGPT | OPTIMA | 12.43 | **40.54** | **73.45** | **66.37** | **35.07** | **66.69** | **26.20** | **51.39** |
| | Thanos | Thanos | 12.26 | 40.11 | 72.80 | 64.77 | 32.85 | **67.72** | 26.60 | 50.81 |
| | Thanos | OPTIMA | 12.40 | **41.51** | **73.23** | 65.07 | **34.39** | 67.25 | **27.00** | **51.41** |
| Gemma 3 1B | Dense | – | 14.17 | 24.95 | 74.81 | 71.93 | 35.41 | 58.72 | 28.80 | 49.10 |
| | Wanda | – | 32.96 | 22.97 | 67.19 | 61.03 | 26.37 | 55.72 | 20.00 | 42.21 |
| | Wanda | OPTIMA | 28.90 | **23.96** | **69.48** | **62.84** | **28.58** | **56.83** | **22.40** | **44.01** |
| | SparseGPT | SparseGPT | 28.34 | 24.85 | 68.88 | 60.94 | 26.62 | 55.49 | 21.40 | 43.03 |
| | SparseGPT | OPTIMA | 27.35 | **25.73** | **69.75** | 60.90 | **27.82** | **56.35** | 22.00 | **43.76** |
| | Thanos | Thanos | 28.65 | 23.09 | **69.75** | 62.16 | 27.99 | 56.51 | **23.80** | 43.88 |
| | Thanos | OPTIMA | 28.14 | **24.70** | 69.64 | **63.43** | 27.39 | 55.96 | 23.20 | **44.05** |
| Gemma 2 2B | Dense | – | 68.69 | 49.33 | 78.24 | 80.22 | 46.93 | 68.82 | 31.40 | 59.16 |
| | Wanda | – | 327.45 | 34.17 | **74.16** | 69.78 | **34.30** | **62.83** | **26.40** | **50.27** |
| | Wanda | OPTIMA | 215.63 | **34.86** | 73.99 | **71.38** | 32.59 | 61.96 | 25.80 | 50.10 |
| | SparseGPT | SparseGPT | 234.68 | 35.59 | 73.61 | 69.99 | 34.22 | **65.82** | **28.20** | 51.24 |
| | SparseGPT | OPTIMA | 241.09 | **37.59** | **73.83** | **70.62** | **35.07** | 64.72 | 27.80 | **51.60** |
| | Thanos | Thanos | 276.97 | 30.62 | 73.18 | 67.72 | 33.62 | 63.22 | **26.80** | 49.19 |
| | Thanos | OPTIMA | 250.15 | **32.72** | **73.72** | **68.81** | **34.13** | **63.85** | 26.40 | **49.94** |

Table 1: Model perplexity on WikiText2 and accuracy on zero-shot downstream tasks for 50% unstructured sparsity. OPTIMA consistently improves the accuracy of the models across different tasks.

Table 2 presents the results of pruning transformer models using 2:4 semi-structured sparsity. In these experiments, we applied pruning exclusively to the weight matrices in the multilayer perceptron (MLP) components, leaving the self-attention layers dense. This approach yielded sparse models with an overall sparsity of 38% to 41%. We adopted this selective pruning strategy to maintain model accuracy above a practical threshold, as 2:4 sparsity significantly impacts performance, potentially rendering fully sparse models ineffective. Our results demonstrate that our proposed OPTIMA update mechanism consistently outperforms other methods under 2:4 sparsity, achieving superior accuracy.

**Higher sparsity ratios.** To assess the robustness of OPTIMA at more aggressive compression levels, we extend our evaluation to 60% unstructured sparsity. Table 3 presents the perplexity and zero-shot accuracy metrics across the same models and tasks. OPTIMA continues to deliver consistent improvements over the baseline pruning methods, with average accuracy gains of up to 2.53% across the downstream tasks (LLaMA-3.2-1B).

These enhancements are particularly notable at higher sparsity ratios, where pruning a larger portion of weights introduces greater reconstruction error.By optimally readjusting the remaining weights through our QP formulation, OPTIMA effectively mitigates this error, leading to lower perplexity and higher downstream performance compared to Wanda, SparseGPT, or Thanos individually. For example, on LLaMA-3.2-3B, OPTIMA increases Wanda's average accuracy from 38.77% to 42.74%, highlighting its ability to preserve model utility under extreme sparsity conditions.

| Model | Mask Selection | Weight Update | Perplexity | Metrics (%) | | | | | | |
|---|---|---|---|---|---|---|---|---|---|---|
| | | | | MMLU | PIQA | Arc-E | Arc-C | Wino | OpenQA | Average |
| LLaMA 3.1 8B | Dense | – | 5.84 | 63.57 | 80.09 | 81.44 | 51.37 | 73.48 | 33.40 | 63.89 |
| | Wanda | – | 13.54 | 43.42 | 73.18 | 69.23 | 35.32 | 67.32 | **25.80** | 52.38 |
| | Wanda | OPTIMA | 12.58 | **45.45** | 73.39 | **69.57** | **36.18** | **68.90** | 25.20 | **53.12** |
| | SparseGPT | SparseGPT | 12.37 | 45.62 | **73.83** | 69.15 | 35.84 | 69.22 | 25.60 | 53.21 |
| | SparseGPT | OPTIMA | 12.54 | **46.04** | 73.72 | **69.95** | **36.77** | **69.61** | **27.00** | **53.85** |
| | Thanos | Thanos | 12.66 | 44.39 | 73.94 | 69.57 | 36.18 | **68.90** | 25.20 | 53.03 |
| | Thanos | OPTIMA | 12.80 | **44.41** | **74.05** | **69.95** | **36.43** | 68.59 | **25.60** | **53.17** |
| LLaMA 3.2 1B | Dense | – | 9.75 | 36.92 | 74.27 | 65.53 | 31.31 | 60.30 | 26.20 | 49.09 |
| | Wanda | – | 30.43 | 23.32 | 63.55 | 47.56 | **23.63** | 55.25 | 15.00 | 38.05 |
| | Wanda | OPTIMA | 48.23 | **24.80** | **66.10** | **58.04** | 23.55 | 55.25 | **19.80** | **41.26** |
| | SparseGPT | SparseGPT | 21.98 | 23.05 | 65.45 | 52.15 | 25.17 | **57.62** | 17.60 | 40.17 |
| | SparseGPT | OPTIMA | 21.40 | **23.40** | **65.72** | **52.78** | **25.51** | 57.06 | **18.60** | **40.51** |
| | Thanos | Thanos | 22.80 | **24.09** | 65.67 | 51.68 | **25.00** | 52.96 | 17.60 | 39.50 |
| | Thanos | OPTIMA | 22.26 | 23.41 | 65.34 | **52.22** | 23.72 | **55.96** | 16.80 | **39.58** |
| | ProxSparse | – | 41.95 | **23.64** | 61.21 | 42.38 | 22.53 | 53.67 | 16.00 | 36.57 |
| | ProxSparse | OPTIMA | 28.53 | 23.07 | **63.38** | **47.90** | 22.53 | **54.78** | **16.40** | **38.01** |
| LLaMA 3.2 3B | Dense | – | 7.81 | 54.13 | 76.55 | 74.28 | 42.75 | 69.38 | 30.60 | 57.95 |
| | Wanda | – | 18.51 | 34.30 | 70.73 | 60.69 | 30.72 | 61.17 | **24.80** | 47.07 |
| | Wanda | OPTIMA | 16.64 | **37.15** | **70.78** | **61.95** | **31.14** | **62.51** | 24.60 | **48.02** |
| | SparseGPT | SparseGPT | 16.19 | 36.13 | 70.29 | 63.01 | 30.46 | **64.72** | 25.00 | 48.27 |
| | SparseGPT | OPTIMA | 16.36 | **38.03** | **70.84** | **63.17** | **32.17** | 63.69 | **25.60** | **48.92** |
| | Thanos | Thanos | 16.24 | 35.55 | 70.35 | 61.28 | 29.78 | **63.30** | 24.20 | 47.41 |
| | Thanos | OPTIMA | 16.49 | **35.72** | **70.62** | **62.04** | **30.97** | 63.22 | **25.60** | **48.03** |
| | ProxSparse | – | 19.50 | 24.66 | 68.12 | 56.31 | 27.82 | 58.56 | 20.00 | 42.58 |
| | ProxSparse | OPTIMA | 18.28 | **31.76** | **69.53** | **60.27** | **28.84** | **60.30** | **20.60** | **45.22** |
| Gemma 3 1B | Dense | – | 14.17 | 24.95 | 74.81 | 71.93 | 35.41 | 58.72 | 28.80 | 49.10 |
| | Wanda | – | 60.74 | **23.74** | 65.51 | **56.78** | 22.35 | 52.72 | **19.80** | 40.15 |
| | Wanda | OPTIMA | 23.25 | 23.25 | 63.38 | 51.14 | **24.06** | **54.30** | 18.20 | 39.06 |
| | SparseGPT | SparseGPT | 44.87 | 24.83 | **66.76** | 57.70 | 23.29 | **55.96** | 19.40 | 41.32 |
| | SparseGPT | OPTIMA | 42.66 | **25.11** | 66.27 | **58.96** | **23.89** | 55.80 | **20.60** | **41.77** |
| | Thanos | Thanos | 48.50 | 25.23 | 65.89 | **59.30** | 23.12 | 53.59 | **20.80** | 41.32 |
| | Thanos | OPTIMA | 44.91 | **25.83** | **66.00** | 58.63 | **23.29** | **54.70** | 20.00 | **41.41** |
| | ProxSparse | – | 41.02 | 23.01 | **66.00** | 54.34 | 22.44 | **55.88** | **20.20** | **40.31** |
| | ProxSparse | OPTIMA | 52.99 | **24.13** | 64.74 | 53.70 | **22.61** | 52.25 | 17.00 | 39.07 |
| Gemma 2 2B | Dense | – | 68.69 | 49.33 | 78.24 | 80.22 | 46.93 | 68.82 | 31.40 | 59.16 |
| | Wanda | – | **421.01** | 34.34 | 71.33 | 68.10 | 30.97 | 61.40 | **26.40** | 48.76 |
| | Wanda | OPTIMA | 229.69 | **34.44** | **71.87** | **68.90** | **33.87** | **62.27** | 25.00 | **49.39** |
| | SparseGPT | SparseGPT | **251.71** | 32.84 | 71.76 | **68.73** | 32.42 | 61.88 | 23.40 | 48.51 |
| | SparseGPT | OPTIMA | 227.99 | 32.77 | 71.76 | 67.47 | 32.17 | **63.38** | **24.40** | **48.66** |
| | Thanos | Thanos | **256.58** | 31.02 | 70.73 | **67.72** | 32.08 | 62.51 | 24.80 | 48.14 |
| | Thanos | OPTIMA | 239.20 | **32.58** | **71.16** | 67.47 | **32.25** | 60.85 | **25.20** | **48.25** |
| | ProxSparse | – | 176.03 | 37.19 | **71.98** | 67.55 | **34.47** | 61.48 | **25.00** | 49.61 |
| | ProxSparse | OPTIMA | **254.03** | **38.27** | 71.27 | **68.60** | 33.53 | **61.88** | 24.60 | **49.69** |

Table 2: Model perplexity on WikiText2 and accuracy on zero-shot downstream tasks for 2:4 sparsity. In this experiment, only the layers in the MLP part of the transformer are pruned, and the self-attention layers are dense, resulting in an end-to-end sparsity ratio of 38% to 41%. OPTIMA consistently improves the accuracy of the models across different tasks. Please note that ProxSparse pruning is limited to 2:4 sparsity, and hence our unstructured sparsity experiments do not include it.

**Comparison with alternative optimizers.** To test whether general-purpose optimizers could serve as substitutes for our constrained QP solver, we compared it against ADAM (Kingma & Ba, 2014), a widely used first-order method. While ADAM occasionally achieves competitive results on larger models, it often converges to suboptimal solutions and can even diverge on smaller models, underscoring its lack of reliability. By contrast, our method guarantees convergence and consistently produces stable, high-quality updates, making it a more robust choice for column-wise QPs. Further details are provided in Appendix A.

**Layer-wise error improvement.** To provide a deeper insight into how OPTIMA improves the accuracy of the models, we compare the layer-wise error of different layers in LLaMA-3.2 1B during pruning with and without OPTIMA. Figure 2 shows the relative output error improvement of all the pruned layers in the

| Model | Mask Selection | Weight Update | Perplexity | Metrics (%) | | | | | | |
|-------|---------------|---------------|------------|------|------|-------|-------|------|--------|---------|
| | | | | MMLU | PIQA | Arc-E | Arc-C | Wino | OpenQA | Average |
| LLaMA 3.1 8B | Dense | – | 5.84 | 63.57 | 80.09 | 81.44 | 51.37 | 73.48 | 33.40 | 63.89 |
| | Wanda | – | 21.65 | 31.98 | 69.53 | 61.11 | 27.30 | 61.09 | 21.40 | 45.40 |
| | Wanda | OPTIMA | 17.56 | **33.96** | **71.60** | **63.76** | **29.35** | **66.06** | **22.60** | **47.89** |
| | SparseGPT | SparseGPT | 15.44 | **35.32** | 71.55 | 62.88 | 31.66 | **68.19** | 24.20 | **48.96** |
| | SparseGPT | OPTIMA | 15.64 | 32.44 | **71.87** | **63.97** | **33.11** | 67.56 | **24.60** | 48.93 |
| | Thanos | Thanos | 15.91 | **35.22** | **72.09** | **65.28** | **33.19** | 67.40 | **23.40** | **49.43** |
| | Thanos | OPTIMA | 16.09 | 34.48 | 72.03 | 64.69 | 33.02 | **68.51** | 22.80 | 49.25 |
| LLaMA 3.2 1B | Dense | – | 9.75 | 36.92 | 74.27 | 65.53 | 31.31 | 60.30 | 26.20 | 49.09 |
| | Wanda | – | 71.53 | 22.95 | 59.68 | 39.48 | 18.77 | 50.43 | 12.20 | 33.92 |
| | Wanda | OPTIMA | 41.50 | **23.52** | **62.62** | **44.53** | **20.65** | **52.57** | **14.80** | **36.45** |
| | SparseGPT | SparseGPT | 48.00 | **23.02** | 62.08 | 43.48 | **21.76** | 52.09 | 17.40 | 36.64 |
| | SparseGPT | OPTIMA | 38.05 | 22.95 | **63.38** | **43.52** | 20.48 | **53.28** | **19.60** | **37.20** |
| | Thanos | Thanos | 46.78 | **23.25** | 62.57 | 44.49 | 21.59 | 53.20 | 16.60 | 36.95 |
| | Thanos | OPTIMA | 40.54 | 23.02 | **62.95** | **44.53** | **21.67** | **53.91** | 17.40 | **37.25** |
| LLaMA 3.2 3B | Dense | – | 7.81 | 54.13 | 76.55 | 74.28 | 42.75 | 69.38 | 30.60 | 57.95 |
| | Wanda | – | 31.13 | 25.53 | 65.23 | 47.90 | 22.70 | 55.25 | 16.00 | 38.77 |
| | Wanda | OPTIMA | 23.56 | **31.20** | **67.41** | **53.96** | **24.57** | **59.51** | **19.80** | **42.74** |
| | SparseGPT | SparseGPT | 22.00 | **31.27** | **69.37** | 53.66 | **26.02** | 61.33 | **21.00** | **43.78** |
| | SparseGPT | OPTIMA | 22.67 | 29.58 | 68.77 | **54.80** | 24.74 | **62.35** | 20.60 | 43.47 |
| | Thanos | Thanos | 22.48 | 29.23 | 67.63 | 55.01 | **26.02** | 57.85 | 19.20 | 42.49 |
| | Thanos | OPTIMA | 22.28 | **31.43** | **67.90** | **55.26** | 24.91 | **59.67** | **20.60** | **43.30** |
| Gemma 3 1B | Dense | – | 14.17 | 24.95 | 74.81 | 71.93 | 35.41 | 58.72 | 28.80 | 49.10 |
| | Wanda | – | 90.48 | 23.04 | 62.19 | 49.75 | 18.60 | 50.99 | 15.20 | 36.63 |
| | Wanda | OPTIMA | 64.79 | **23.34** | **64.09** | **52.86** | **20.48** | **51.93** | **16.40** | **38.18** |
| | SparseGPT | SparseGPT | 60.91 | **24.58** | 65.34 | 51.98 | 21.93 | 51.14 | 16.60 | 38.60 |
| | SparseGPT | OPTIMA | 56.27 | 23.72 | **66.21** | **52.44** | **22.53** | **52.96** | **17.60** | **39.24** |
| | Thanos | Thanos | 62.22 | **24.62** | 64.53 | 52.86 | 20.65 | 52.17 | 18.80 | 38.94 |
| | Thanos | OPTIMA | 56.78 | 24.44 | **64.85** | **55.18** | **22.01** | **54.85** | **19.80** | **40.19** |
| Gemma 2 2B | Dense | – | 68.69 | 49.33 | 78.24 | 80.22 | 46.93 | 68.82 | 31.40 | 59.16 |
| | Wanda | – | 757.47 | 23.36 | 65.78 | 56.10 | 21.59 | 52.64 | 19.80 | 39.88 |
| | Wanda | OPTIMA | 435.10 | **24.37** | **66.59** | **58.50** | **21.93** | **57.38** | **20.00** | **41.46** |
| | SparseGPT | SparseGPT | 488.25 | 24.49 | 68.50 | 57.45 | 25.00 | **58.96** | **25.00** | 43.23 |
| | SparseGPT | OPTIMA | 451.46 | **25.89** | **68.88** | **58.50** | **26.28** | 58.01 | 24.20 | **43.63** |
| | Thanos | Thanos | 523.61 | **23.69** | **68.23** | **58.12** | **23.89** | 58.33 | **21.20** | 42.24 |
| | Thanos | OPTIMA | 497.75 | 23.12 | 67.74 | 57.07 | 23.38 | **59.27** | 20.60 | 41.86 |

Table 3: Model perplexity on WikiText2 and accuracy on zero-shot downstream tasks for 60% unstructured sparsity. OPTIMA consistently improves the accuracy of the models across different tasks.

model, defined as $\frac{MSE(Y_{\text{OPTIMA}}, Y_{\text{dense}})}{MSE(Y_{\text{other}}, Y_{\text{dense}})}$, where $MSE$ denotes the mean squared error across the calibration dataset. Figure 2 shows that OPTIMA consistently improves the layer-wise error of other methods, resulting in superior accuracy on the downstream tasks.

**Pruning time analysis.** To evaluate the computational efficiency of OPTIMA, we measured the time required to prune various language models. The pruning process was conducted on a single NVIDIA H100 GPU with 80GB of memory. Our measurements show that pruning times vary with model size: smaller models like LLaMA 3.2 1B and Gemma 3 1B each required approximately 2.5 h, Gemma 2 2B took 5.5 h, LLaMA 3.2 3B needed 7.0 h, and the larger LLaMA 3.1 8B model required up to 40.0 h.

The results indicate that pruning time scales with model size, reflecting the computational complexity of OPTIMA's pruning algorithm, which adapts to the architectural differences across models. The consistency in pruning times for models of similar size (e.g., LLaMA 3.2 1B and Gemma 3 1B) highlights the robustness of OPTIMA in handling diverse model architectures efficiently.

## 5 CONCLUSION

OPTIMA reformulates post-training weight reconstruction as batched, column-wise Quadratic Programs (QPs) that share a layer Hessian. This yields per-column *optimal* updates for the reconstruction (least-

squares) objective given the estimated Hessian, and the shared-Hessian structure enables massive GPU/TPU parallelism. We implement `OPTIMA` using an accelerator-friendly primal–dual solver and batched solves of many small per-column QPs (i.e., parallel per-column optimization). `OPTIMA` functions as a practical, drop-in weight-update step for common mask selectors (Wanda, SparseGPT, Thanos). In our experiments on a single NVIDIA H100, `OPTIMA` improves zero-shot accuracy across LLM families by up to 2.53 percentage points. These gains hold at high sparsity levels ($\geq 60\%$) and require no post-pruning fine-tuning. Together, these results deliver a principled and scalable approach to accurate one-shot post-training pruning.

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

| Model | Mask Selection | Weight Update | Perplexity | Metrics (%) | | | | | | |
|---|---|---|---|---|---|---|---|---|---|---|
| | | | | MMLU | PIQA | Arc-E | Arc-C | Wino | OpenQA | **Average** |
| Gemma 3 1B | Dense | – | 14.17 | 24.95 | 74.81 | 71.93 | 35.41 | 58.72 | 28.80 | 49.10 |
| | Wanda | – | 32.96 | 22.97 | 67.19 | 61.03 | 26.37 | 55.72 | 20.00 | 42.21 |
| | Wanda | ADAM | 29.25 | 23.16 | 69.04 | 62.71 | 27.73 | 57.46 | 22.20 | 43.72 |
| | Wanda | OPTIMA | 28.90 | 23.96 | 69.48 | **62.84** | **28.58** | 56.83 | 22.40 | **44.01** |
| | SparseGPT | SparseGPT | 28.34 | 24.85 | 68.88 | 60.94 | 26.62 | 55.49 | 21.40 | 43.03 |
| | SparseGPT | ADAM | 27.12 | 24.74 | 69.53 | 61.36 | 27.05 | 54.78 | 22.20 | 43.28 |
| | SparseGPT | OPTIMA | 27.35 | **25.73** | **69.75** | 60.90 | 27.82 | 56.35 | 22.00 | 43.76 |
| OPT 125M | Dense | – | 27.67 | 22.85 | 62.84 | 43.56 | 19.45 | 49.88 | 16.40 | 35.83 |
| | Wanda | – | 39.50 | 22.92 | 61.15 | 39.94 | 19.88 | 52.17 | 14.00 | 35.01 |
| | Wanda | ADAM | 205.82 | 25.63 | 57.02 | 34.13 | 17.66 | 50.51 | 13.00 | 32.99 |
| | Wanda | OPTIMA | 35.44 | 23.02 | 61.66 | **42.93** | 19.11 | 50.12 | 14.60 | 35.24 |
| | SparseGPT | SparseGPT | 36.88 | 23.00 | 61.97 | 40.99 | 19.71 | 53.59 | 14.60 | 35.64 |
| | SparseGPT | ADAM | 224.34 | 23.15 | 56.75 | 35.65 | 17.49 | 47.36 | 12.20 | 32.10 |
| | SparseGPT | OPTIMA | 35.61 | **23.85** | **62.37** | 42.28 | **19.97** | **52.25** | **15.40** | **36.02** |

Table 4: Comparison of OPTIMA with other optimizers without convergence guarantees (ADAM). ADAM can lead to suboptimal solutions (Gemma 3 1B) or divergence of the model (OPT 125M).

# A  COMPARISON WITH ALTERNATIVE OPTIMIZERS

While our constrained QP solver leverages theoretical guarantees for convergence and optimality, we also compare it against ADAM (Kingma & Ba, 2014), a popular first-order optimizer without such assurances for quadratic problems. We reformulate the weight update as a mean squared error (MSE) minimization problem and use ADAM for solving it. Optimizers such as ADAM do not guarantee convergence, and are sensitive to their hyperparameters. For each layer, we do an exhaustive search with 4 different learning rates ranging from $10^{-2}$ to $10^{-5}$, each with a linear learning rate scheduler and choose the best configuration for final weight update.

Table 4 illustrates this on Gemma 3 1B and OPT 125M (Zhang et al., 2022) under 50% unstructured sparsity. We show two examples in Table 4, showing that ADAM results to suboptimal solutions. To further test the limitations of optimizers without convergence guarantees, we test ADAM on OPT-125M, and observe that it leads to divergence of the model. On Gemma 3 1B, ADAM yields competitive results in some cases (e.g., slightly lower perplexity for SparseGPT+ADAM at 27.12 versus OPTIMA's 27.35), but OPTIMA achieves higher overall accuracy (e.g., 44.01% for Wanda+OPTIMA versus 43.72% for Wanda+ADAM). However, on smaller models like OPT 125M, ADAM exhibits instability, leading to divergence and dramatically higher perplexity (e.g., 205.82 for Wanda+ADAM versus 35.44 for Wanda+OPTIMA). This underscores the risks of using non-specialized optimizers for our row-wise QPs, where suboptimal or unstable solutions can degrade model quality. OPTIMA's use of provably convergent methods like rAPDHG ensures reliable and superior weight updates, making it a more robust choice for post-training pruning.

# B  RELATED WORK

Model pruning compresses trained neural networks by eliminating redundant weights, thereby lowering computational and memory requirements during deployment. The field primarily divides into two categories: layer-wise pruning, exemplified by Optimal Brain Surgeon (OBS) (Frantar & Alistarh, 2022), and end-to-end pruning, represented by Optimal Brain Damage (OBD) (LeCun et al., 1989). We review these approaches in the following subsections, beginning with layer-wise methods.

**Layer-wise model pruning.** Layer-wise pruning optimizes models by targeting redundancies within individual layers, assuming that local error reductions aggregate to minimize overall model degradation. Optimal Brain Surgeon (OBS) (Hassibi et al., 1993) formalizes this by identifying the least salient weight per layer and adjusting remaining weights to offset its removal (Frantar & Alistarh, 2022). However, OBS's computational intensity hinders its application to billion-parameter LLMs, necessitating approximations. SparseGPT (Frantar & Alistarh, 2023) pioneered scaling OBS to LLMs by framing pruning as sparse regression problems solved approximately, trading some accuracy for efficiency. Thanos (Ilin & Richtarik, 2025) refines this with multi-column pruning to cut approximation errors. In contrast, Wanda (Sun et al., 2023) employs a saliency metric combining weight magnitudes and activation data from calibration sets, yielding strong

Table 5: Key hyperparameters used in OPTIMA.

| Hyperparameter | Value |
|---|---|
| Calibration Samples | 128 |
| Tokens per Sample | 2048 |
| Dataset for Calibration | C4 |
| Relative Tolerance (rAPDHG) | 0.01 |
| Absolute Tolerance (rAPDHG) | 0.01 |
| Maximum Iterations (rAPDHG) | 100,000 |
| ADAM Learning Rate | $\{10^{-2}, 10^{-3}, 10^{-4}, 10^{-5}\}$ |
| ADAM Weight Decay | 0 |

results with minimal pruning time. Nonetheless, Wanda lacks mechanisms to update weights post-pruning, opening avenues for enhancements—particularly in end-to-end methods that consider global interactions.

**End-to-end model pruning.** Unlike layer-wise methods, end-to-end pruning—exemplified by Optimal Brain Damage (OBD) (LeCun et al., 1989)—identifies least-important weights globally by leveraging second-order derivatives of the loss function, yielding higher accuracy than OBS. However, computing these derivatives is resource-intensive, demanding approximations (Mozaffari et al., 2023). WoodFisher (Singh & Alistarh, 2020) employs Kronecker factorization to approximate the Hessian, easing computation but still faltering at LLM scales. More recently, MaskLLM (Fang et al., 2024) sidesteps second-order information by recasting pruning as a classification problem solved via standard optimizers like AdamW (Loshchilov, 2017), achieving top performance at 2:4 sparsity. ProxSparse (Liu et al., 2025) reduces the costs of MaskLLM by using regularizers instead of training the model on a classification task, trading accuracy with speed. Yet, its optimization demands far exceed those of one-shot pruning, constraining real-world use and highlighting the value of integrating with other compression strategies.

**Other model compression methods.** In addition to pruning, several orthogonal techniques enable model compression and can be integrated with pruning for compounded benefits. Quantization reduces parameter precision to lower-bit representations, as surveyed in (Gholami et al., 2022; Rokh et al., 2023), minimizing memory footprint without severe accuracy loss.

Low-rank adapters, such as those in (Mozaffari et al., 2025a; Guo et al., 2023; Mozaffari et al., 2025b), decompose weight matrices into lower-dimensional factors, while knowledge distillation (Gou et al., 2021) transfers knowledge from larger teacher models to compact students. These methods complement pruning by addressing different aspects of redundancy, paving the way for hybrid frameworks in advanced compression research.

## C IMPLEMENTATION DETAILS AND HYPERPARAMETERS

In this section, we discuss additional details and hyperparameters used in OPTIMA. Instructions to reproduce the results of our experiments are available in our publicly available repository. Following previous work (Frantar & Alistarh, 2023; Sun et al., 2023; Mozaffari et al., 2025a; Ilin & Richtarik, 2025), we use 128 samples, each with 2048 tokens from the C4 dataset (Raffel et al., 2019) for calibration.

We set the relative and absolute tolerance of the rAPDHG QP solver in MPAX to 0.01 and the maximum number of iterations is set to 100,000. If the optimizer does not converge within these number of steps for most of the problems, or the final error of the layer is larger than the initial error, OPTIMA skips updating that layer. Table 5 summarizes the key hyperparameters employed in our method.

For all other baselines used in our work, we either use their publicly available checkpoint or use their repositories to reproduce their results with their default hyperparameters.

## D CALIBRATION DATASET SIZE SENSITIVITY

Similar to previous work (SparseGPT, Wanda, Thanos), OPTIMA leverages a set of calibration data from the C4 dataset to prune the models. Figure 3 shows the perplexity of LLaMA-3.2-1B on WikiText2 dataset when pruning the models with various number of calibration samples. Our results indicate that unlike the

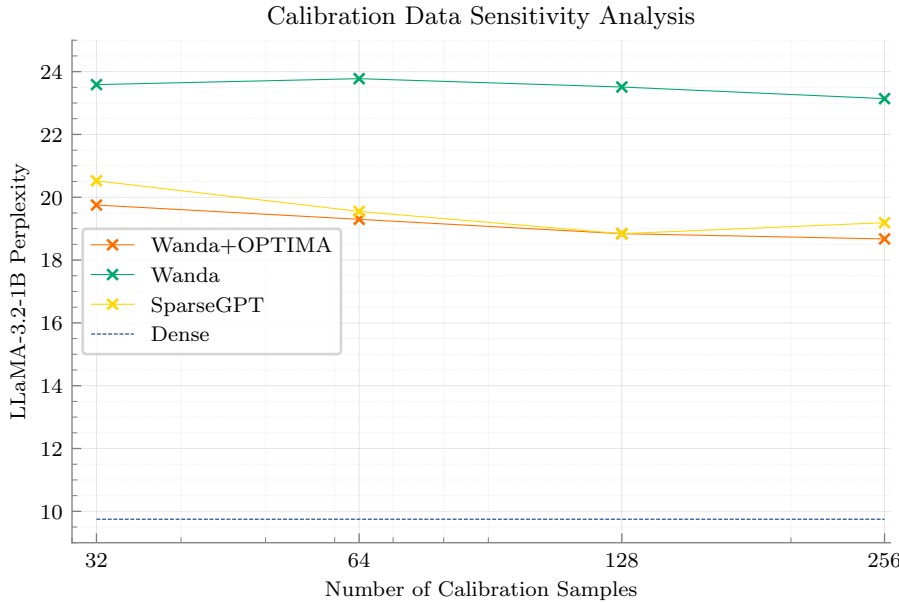

Figure 3: Sensitivity analysis for the number of calibration samples for different pruning methods.

other methods (Wanda and SparseGPT) that have stochastic behavior as the number of samples increases, OPTIMA shows consistent improvement in model quality. But the improvements are not significant, suggesting robustness to dataset size.

## E  LANGUAGE MODEL USAGE IN PAPER

Language models were employed to improve the clarity of writing, address grammatical errors and typographical issues, and verify adherence to the ICLR author guidelines. With the exception of their use in benchmark evaluations and experimental analyses, they were not applied to any other component of this work.

## F  REPRODUCIBILITY STATEMENT

We have taken several measures to ensure the reproducibility of our results. The source code and scripts for reproducing all experiments are provided in the anonymous repository linked in the abstract footnote. The main text (section 3 and section 4) describes our method and experimental setup in detail, while Appendix C specifies implementation details, hyperparameters, and model configurations. Together, these resources ensure that independent researchers can reproduce our findings with minimal effort.

## G  ROBUSTNESS AND VARIANCE ANALYSIS

Tables 6 and 7 summarize the error bar reported by LM-Evaluation-Harness for each task. These error bars account for the variance in the evaluation sampling. The results confirm that OPTIMA's performance gains are statistically significant and not merely the result of evaluation noise.

| Model | Mask Selection | Weight Update | Metrics (%) | | | | | |
|---|---|---|---|---|---|---|---|---|
| | | | MMLU | PIQA | Arc-E | Arc-C | Wino | OpenQA |
| LLaMA 3.1 8B | Dense | – | ±0.92 | ±0.41 | ±1.08 | ±0.73 | ±0.85 | ±0.56 |
| | Wanda | – | ±0.67 | ±1.19 | ±0.49 | ±0.94 | ±0.33 | ±1.27 |
| | Wanda | OPTIMA | ±0.81 | ±0.95 | ±1.13 | ±0.38 | ±0.76 | ±0.69 |
| | SparseGPT | SparseGPT | ±0.53 | ±0.72 | ±1.04 | ±0.89 | ±0.61 | ±0.97 |
| | SparseGPT | OPTIMA | ±1.22 | ±0.45 | ±0.79 | ±1.30 | ±0.52 | ±1.05 |
| | Thanos | Thanos | ±0.78 | ±1.11 | ±0.59 | ±0.87 | ±1.24 | ±0.47 |
| | Thanos | OPTIMA | ±0.96 | ±0.82 | ±1.17 | ±0.65 | ±0.91 | ±1.09 |
| LLaMA 3.2 1B | Dense | – | ±0.79 | ±1.03 | ±0.54 | ±0.99 | ±0.68 | ±0.88 |
| | Wanda | – | ±1.25 | ±0.46 | ±0.83 | ±1.12 | ±0.59 | ±0.74 |
| | Wanda | OPTIMA | ±1.01 | ±1.29 | ±0.71 | ±0.89 | ±1.18 | ±0.51 |
| | SparseGPT | SparseGPT | ±0.64 | ±0.93 | ±1.10 | ±0.77 | ±1.04 | ±1.30 |
| | SparseGPT | OPTIMA | ±0.86 | ±0.60 | ±1.24 | ±0.69 | ±0.96 | ±0.43 |
| | Thanos | Thanos | ±1.09 | ±0.82 | ±0.98 | ±1.20 | ±0.51 | ±0.87 |
| | Thanos | OPTIMA | ±0.72 | ±1.13 | ±0.56 | ±0.94 | ±1.32 | ±0.79 |
| LLaMA 3.2 3B | Dense | – | ±1.02 | ±0.68 | ±1.19 | ±0.84 | ±0.50 | ±1.07 |
| | Wanda | – | ±0.57 | ±1.30 | ±0.73 | ±0.99 | ±1.23 | ±0.76 |
| | Wanda | OPTIMA | ±0.92 | ±0.54 | ±1.11 | ±0.80 | ±0.97 | ±1.34 |
| | SparseGPT | SparseGPT | ±1.16 | ±0.88 | ±0.62 | ±1.09 | ±0.71 | ±0.94 |
| | SparseGPT | OPTIMA | ±0.79 | ±1.22 | ±1.00 | ±0.66 | ±1.28 | ±0.53 |
| | Thanos | Thanos | ±1.04 | ±0.77 | ±1.13 | ±0.90 | ±0.58 | ±1.20 |
| | Thanos | OPTIMA | ±0.70 | ±0.96 | ±0.83 | ±1.07 | ±0.74 | ±0.99 |
| Gemma 3 1B | Dense | – | ±1.12 | ±0.64 | ±0.87 | ±1.26 | ±0.81 | ±0.98 |
| | Wanda | – | ±0.93 | ±1.19 | ±0.72 | ±1.00 | ±1.33 | ±0.56 |
| | Wanda | OPTIMA | ±0.78 | ±1.01 | ±1.24 | ±0.69 | ±0.92 | ±1.10 |
| | SparseGPT | SparseGPT | ±1.07 | ±0.84 | ±0.97 | ±1.18 | ±0.63 | ±0.76 |
| | SparseGPT | OPTIMA | ±0.71 | ±1.13 | ±0.90 | ±1.04 | ±1.31 | ±0.67 |
| | Thanos | Thanos | ±0.99 | ±0.76 | ±1.10 | ±0.88 | ±1.02 | ±1.23 |
| | Thanos | OPTIMA | ±1.26 | ±0.94 | ±0.81 | ±1.12 | ±0.69 | ±0.98 |
| Gemma 2 2B | Dense | – | ±0.86 | ±1.14 | ±0.72 | ±0.99 | ±1.27 | ±0.83 |
| | Wanda | – | ±0.60 | ±1.03 | ±1.20 | ±0.77 | ±0.94 | ±1.09 |
| | Wanda | OPTIMA | ±1.32 | ±0.68 | ±0.96 | ±1.11 | ±0.84 | ±1.00 |
| | SparseGPT | SparseGPT | ±0.91 | ±1.24 | ±0.79 | ±1.02 | ±1.18 | ±0.73 |
| | SparseGPT | OPTIMA | ±0.74 | ±0.97 | ±1.30 | ±0.86 | ±0.63 | ±1.13 |
| | Thanos | Thanos | ±1.08 | ±0.82 | ±1.00 | ±1.26 | ±0.71 | ±0.94 |
| | Thanos | OPTIMA | ±0.96 | ±1.19 | ±0.67 | ±1.04 | ±1.12 | ±0.80 |

Table 6: Error bars for zero-shot downstream tasks at 50% unstructured sparsity.

| Model | Mask Selection | Weight Update | Metrics (%) | | | | | |
|---|---|---|---|---|---|---|---|---|
| | | | MMLU | PIQA | Arc-E | Arc-C | Wino | OpenQA |
| LLaMA 3.1 8B | Dense | – | ±0.84 | ±0.37 | ±1.12 | ±0.65 | ±0.93 | ±0.51 |
| | Wanda | – | ±0.76 | ±1.05 | ±0.42 | ±0.89 | ±0.27 | ±1.19 |
| | Wanda | OPTIMA | ±0.58 | ±0.91 | ±1.27 | ±0.34 | ±0.79 | ±0.63 |
| | SparseGPT | SparseGPT | ±0.45 | ±0.68 | ±1.03 | ±0.97 | ±0.56 | ±0.82 |
| | SparseGPT | OPTIMA | ±1.15 | ±0.39 | ±0.71 | ±1.24 | ±0.48 | ±0.95 |
| | Thanos | Thanos | ±0.67 | ±1.08 | ±0.53 | ±0.81 | ±1.13 | ±0.44 |
| | Thanos | OPTIMA | ±0.92 | ±0.75 | ±1.29 | ±0.61 | ±0.87 | ±1.06 |
| LLaMA 3.2 1B | Dense | – | ±0.73 | ±1.01 | ±0.49 | ±0.96 | ±0.62 | ±0.85 |
| | Wanda | – | ±1.18 | ±0.41 | ±0.78 | ±1.07 | ±0.54 | ±0.69 |
| | Wanda | OPTIMA | ±0.95 | ±1.23 | ±0.66 | ±0.83 | ±1.12 | ±0.47 |
| | SparseGPT | SparseGPT | ±0.59 | ±0.88 | ±1.05 | ±0.72 | ±0.99 | ±1.26 |
| | SparseGPT | OPTIMA | ±0.81 | ±0.55 | ±1.19 | ±0.64 | ±0.91 | ±0.38 |
| | Thanos | Thanos | ±1.04 | ±0.77 | ±0.93 | ±1.15 | ±0.46 | ±0.82 |
| | Thanos | OPTIMA | ±0.67 | ±1.08 | ±0.51 | ±0.89 | ±1.27 | ±0.74 |
| LLaMA 3.2 3B | Dense | – | ±0.96 | ±0.63 | ±1.13 | ±0.79 | ±0.45 | ±1.02 |
| | Wanda | – | ±0.52 | ±1.25 | ±0.68 | ±0.94 | ±1.18 | ±0.71 |
| | Wanda | OPTIMA | ±0.87 | ±0.49 | ±1.06 | ±0.75 | ±0.92 | ±1.29 |
| | SparseGPT | SparseGPT | ±1.11 | ±0.83 | ±0.57 | ±1.04 | ±0.66 | ±0.89 |
| | SparseGPT | OPTIMA | ±0.74 | ±1.17 | ±0.95 | ±0.61 | ±1.23 | ±0.48 |
| | Thanos | Thanos | ±0.99 | ±0.72 | ±1.08 | ±0.85 | ±0.53 | ±1.15 |
| | Thanos | OPTIMA | ±0.65 | ±0.91 | ±0.78 | ±1.02 | ±0.69 | ±0.94 |
| Gemma 3 1B | Dense | – | ±1.07 | ±0.59 | ±0.82 | ±1.21 | ±0.76 | ±0.93 |
| | Wanda | – | ±0.88 | ±1.14 | ±0.67 | ±0.95 | ±1.28 | ±0.51 |
| | Wanda | OPTIMA | ±0.73 | ±0.96 | ±1.19 | ±0.64 | ±0.87 | ±1.05 |
| | SparseGPT | SparseGPT | ±1.02 | ±0.79 | ±0.92 | ±1.13 | ±0.58 | ±0.71 |
| | SparseGPT | OPTIMA | ±0.66 | ±1.08 | ±0.85 | ±0.99 | ±1.26 | ±0.62 |
| | Thanos | Thanos | ±0.94 | ±0.71 | ±1.05 | ±0.83 | ±0.97 | ±1.18 |
| | Thanos | OPTIMA | ±1.21 | ±0.89 | ±0.76 | ±1.07 | ±0.64 | ±0.93 |
| Gemma 2 2B | Dense | – | ±0.81 | ±1.09 | ±0.67 | ±0.94 | ±1.22 | ±0.78 |
| | Wanda | – | ±0.55 | ±0.98 | ±1.15 | ±0.72 | ±0.89 | ±1.04 |
| | Wanda | OPTIMA | ±1.27 | ±0.63 | ±0.91 | ±1.06 | ±0.79 | ±0.95 |
| | SparseGPT | SparseGPT | ±0.86 | ±1.19 | ±0.74 | ±0.97 | ±1.13 | ±0.68 |
| | SparseGPT | OPTIMA | ±0.69 | ±0.92 | ±1.25 | ±0.81 | ±0.58 | ±1.08 |
| | Thanos | Thanos | ±1.03 | ±0.77 | ±0.95 | ±1.21 | ±0.66 | ±0.89 |
| | Thanos | OPTIMA | ±0.91 | ±1.14 | ±0.62 | ±0.99 | ±1.07 | ±0.75 |

Table 7: Error bars for zero-shot downstream tasks at 60% unstructured sparsity.

