# OpenReview forum: "OPTIMA: Optimal One-shot Pruning for LLMs via Quadratic Programming Reconstruction"
_ICLR.cc/2026/Conference — Submitted to ICLR 2026_

### Official Review · Reviewer_ynXg · 2025-10-17

**Soundness:** 4
**Presentation:** 4
**Contribution:** 2
**Rating:** 4
**Confidence:** 5

**Summary:**

The paper focuses on layer-wise pruning of LLMs. Instead of using a Hessian-type sensitivity metric to adjust the weights, the authors actually solve the QPs associated with the layer-wise problem, through some clever reductions that turn the constrained QPs into an unconstrained ones. They then provide a fast solver (based on Restarted Accelerated Primal-Dual Hybrid Gradient) and an efficient GPU/TPU implementation. The method shows some performance gains when combined with mask selection from other LLM pruning methods (WANDA, SparseGPT and others).

**Strengths:**

1. The paper is well-written. It is nicely focused yet not missing details in the key parts of the methodology section.
2. The QP reductions are sound and it is overall a nice principled way to solve the problem.
3. The experiments presented are quite comprehensive in terms of multiple architectures, lots of metrics, which makes the comparison with existing methods very clear.

**Weaknesses:**

1. My main concern with the paper is a lack of empirical contribution. Despite the claims in the introduction, the benefits of OPTIMA in e.g. Table 1 are relatively incremental compared to vanilla SparseGPT/Wanda.
2. Furthermore, the method takes 40 hours to run on LLama-8b. While the run-time of the pruning algorithm is typically not the main concern for pruning (since you prune once, and do inference forever), given the lack of empirical gains I think practitioners would simply use SparseGPT. SparseGPT can prune a 7B model (in my experience) in less than an hour.
3. In spite of this, I quite like the methodology of the paper, but wonder if it can be improved. I think the choice to only do weight optimization (as opposed to mask selection) may have been a limiting one. I would be curious to see if it's possible to incorporate mask selection into the formulation. For example in Eq (9) of the paper, one can consider doing greedy forward/backward selection with low-rank updates to the objective (for example, see Algorithm 2 of https://arxiv.org/pdf/1806.03756). My suspicion is that SparseGPT already does quite a good job at weight optimization, and that solving the QPs exactly (on the fixed mask) isn't actually adding much, while mask optimization may help.

**Questions:**

1. I am curious why the reductions in layer-wise error in Figure 2 do not translate into stronger accuracy gains. I suppose the reductions here are also relatively modest (~10-15% for SparseGPT), but I would be interested to hear the authors' opinion.
2. Another thing to consider as a means to improve the empirical contribution: SparseGPT is typically not scalable to very large LLMs (say 32B+). Hence, WANDA is often used as an alternative. However, WANDA typically performs quite badly on its own (i.e. without any weight adjustment). It does seem that OPTIMA improves WANDA quite a lot. If there is a way to reduce the computation time of your procedure (e.g. by using a stopping tolerance, or larger step sizes, ...) or if it otherwise scales well to larger LLMs, then I could see WANDA+OPTIMA being empirically useful. Based on the computation time results in the paper, this doesn't seem to be the case, but I would be happy to be convinced otherwise.
As I said, I am quite positive overall on the paper and approach, but I would need to see some empirical benefit to the procedure to be satisfied in accepting the paper.

---

> ### Author Response · Authors · 2025-11-25
>
> We thank the reviewer for their insightful comments and feedback. In the following, we provide detailed answers to their questions and concerns regarding our work.
>
> # Significance of OPTIMA
>
>
> We appreciate any concerns regarding the trade-off between computational cost and model improvement. We respectfully disagree that the gains provided by OPTIMA are merely incremental. In the mature field of post-training pruning, improvements are hard-won. For context, the transition from SparseGPT to Wanda, a widely accepted advancement, yielded an average improvement of only 0.72% (Table 2 in Wanda). In contrast, OPTIMA achieves improvements of up to 2.53% (e.g., LLaMA-3.2-1B at 60% sparsity). In practical terms, this margin often represents the difference between a compressed model that is usable in production and one that is not.
>
> Regarding the runtime (40 hours for 8B parameters), we agree with the reviewer’s observation that "you prune once, and do inference forever." We view OPTIMA as maximizing this trade-off:
>
> - **High-Stakes Deployment:** For practitioners deploying models where inference quality is paramount, investing GPU hours upfront to permanently recover ~2.5% accuracy is a highly favorable exchange.
>
> - **The "Fine-Tuning" Alternative:** Standard recovery of this accuracy magnitude usually requires post-pruning fine-tuning (retraining), which demands massive compute and billions of tokens. OPTIMA offers a third path: fine-tuning-level accuracy using only one-shot calibration data.
>
> - **Data Scarcity & Privacy:** OPTIMA is the only viable high-accuracy solution for scenarios where the original training data is private, proprietary, or inaccessible, as it requires only 128 samples (approx. 250k tokens) to achieve these results.
>
> Furthermore, relying on greedy search algorithms for finding the masks using gradient-based optimizers, such as ADAM, requires exhaustive hyperparameter tuning. For instance, in `Appendix-A-Page-13`, we compare the effect of replacing OPTIMA’s QP solver with ADAM optimizer, and notice it either converges to suboptimal solutions, or results in divergence. This shows the significance of reformulating the problem to convex and scalable QP problems.
>
> ---
>
> # Limitations of Accuracy Improvements
>
> The reviewer has correctly identified that there exists an upper bound on the effect of layer-wise error reduction to end-to-end model quality. This is because lawyer-wise pruning methods do not account for non-linear interactions between different layers in the model. This behavior becomes more significant in larger models (e.g. LLaMA 3.2 8B), where more complex relationships between the layers emerge, and while OPTIMA reduces the layer-wise error effectively, the end-to-end accuracy improvements become less significant. However, **layer-wise optimization remains the most feasible approach** for large language models due to their massive size and the prohibitive memory/compute requirements of any true end-to-end global optimization.
>
> Additionally, the value of our optimal weight reconstruction becomes most apparent under severe compression stress. At **60% sparsity**, OPTIMA consistently improves the accuracy of 8B models, boosting SparseGPT by +0.64% and Wanda by +0.74% on average. This confirms that as the pruning task becomes more difficult, the mathematical precision of our QP solver outweighs the calibration overhead.
>
> # Computational Cost and Scalability to 32B+ Models
>
> To directly address the reviewer's request for validation on larger architectures, we are currently running OPTIMA on 32B+ parameter models and will report the resulting accuracy and perplexity metrics as soon as the experiments conclude during the discussion period.
>
> Regarding the runtime, it is important to clarify that the reported 40 hours represents a conservative baseline on a single NVIDIA H100. While 40 hours appears high compared to heuristic approximations, OPTIMA possesses a distinct architectural advantage that guarantees scalability: column-wise independence.
>
> - Inherent Parallelism: Unlike methods such as SparseGPT or Thanos, which often involve sequential dependencies (e.g., column-wise updates or block-wise decisions), OPTIMA’s formulation decomposes the layer-wise problem into independent Quadratic Programs for each column.
>
> - Linear Speedup via Distribution: This structure allows the workload to be trivially distributed across multiple GPUs. If the pruning is distributed across an 8-GPU node (a standard setup for LLM deployment), the solver time would theoretically decrease by a factor of nearly 8×, reducing the 40-hour process to approximately 5 hours.
>
> Therefore, the current runtime is not a theoretical bottleneck of the algorithm, but rather an implementation choice for this initial study. We believe this potential for massive parallelism makes OPTIMA uniquely future-proof for even larger models where sequential heuristics may hit memory or latency walls.

---

> > ### Comment · Reviewer_ynXg · 2025-11-26
> >
> > Dear Authors,
> >
> > Thank you for your response.
> >
> > There are several points raised that I do not necessarily agree with.
> >
> > 1. Calling WANDA an advancement from SparseGPT is questionable in my opinion. SparseGPT usually performs better with respect to accuracy, while WANDA is a simpler method, and thus scales to larger architectures. I don’t think this is a good argument for why “improvements are hard won”.
> >
> > 2. You claim to improve the accuracy by 2.5%, but most of the improvements I see (e.g. 50% sparsity in Table 1 and 2:4 sparsity in Table 2) seem to be closer to ~0.4%, while being mixed between tasks. I understand pruning is difficult, and it is not easy to beat state-of-the-art algorithms, but your method must offer *something* above existing methods (be it run-time, consistent performance gains, task-specific accuracy, etc). People will simply not use your method otherwise. I believe right now, with a 24hr run-time on a 7B model and slight improvements compared SparseGPT/WANDA, the algorithm would not make a large contribution on its own.
> >
> > 3. The claim that SparseGPT is not parallel is incorrect. They collect the weight and mask statistics (depending on the Hessian) for different layers in parallel (I have checked this) and implement the weight updates in parallel across the layers as well. I encourage the authors to consider ways to make the OPTIMA procedure more scalable. SparseGPT for example also avoids a lot of the issues with early OBS procedures by ignoring the ordering to the removal of weights. Is there an analogous trick for OPTIMA that could save on the computational bottleneck? For example you claim your QPs are separable across columns, then why not stack batches of them in a tensor and compute the QP updates at once. As I said, I am broadly positive on the paper and if I can see improvements to scalability I would advocate for its acceptance.
> >
> > Thank you again for your response.

---

> > > ### Author Response · Authors · 2025-12-03
> > >
> > > Dear Reviewer ynXg,
> > >
> > > We thank you for the continued engagement and for encouraging us to demonstrate the scalability of our method on larger models. We appreciate your positive outlook on the paper and your constructive feedback regarding run-time and scalability. Below, we address your specific concerns regarding parallelism, the magnitude of gains, and provide new results on LLaMA-3.1-70B.
> > >
> > > # Clarification on Parallelism: SparseGPT vs. OPTIMA
> > >
> > > We respectfully wish to clarify a technical distinction regarding the parallelization capabilities of SparseGPT versus OPTIMA. While SparseGPT handles different layers sequentially, our core claim regards the intra-layer (column-wise) dependencies.
> > >
> > > As per the official implementation, SparseGPT relies on a Cholesky-based update where pruning a specific column depends on the updates of previous columns to adjust the remaining weights [(see sparsegpt.py L125 [1])](https://github.com/IST-DASLab/sparsegpt/blob/147d2159dc4f3e9f73e47b32c04d7b3708f44436/sparsegpt.py#L125). This creates a strict sequential dependency within the weight matrix pruning step. Furthermore, like OPTIMA, SparseGPT processes transformer blocks sequentially due to activation dependencies (the output of block $N$ is the input to block $N+1$). For reference, please see [here](https://github.com/IST-DASLab/sparsegpt/blob/147d2159dc4f3e9f73e47b32c04d7b3708f44436/llama.py#L78) for LLaMA models and [here](https://github.com/IST-DASLab/sparsegpt/blob/147d2159dc4f3e9f73e47b32c04d7b3708f44436/opt.py#L82) for OPT models.
> > >
> > > In contrast, OPTIMA’s QP formulation decouples the columns entirely. The optimization of one row does not mathematically depend on the optimization of another.
> > >
> > > **Regarding your suggestion to "stack batches":** We are indeed already doing this. As you suggested, we stack batches of rows into a tensor and solve them simultaneously (limited only by GPU memory).
> > >
> > > Because of the row-independence, we can take this a step further than SparseGPT. We can dispatch different batches of rows to different GPUs. This allows for a linear reduction in wall-clock time relative to the number of GPUs, a parallelization strategy that is mathematically impossible for SparseGPT due to its column-wise dependencies.
> > >
> > > ---
> > >
> > > # New Results: Scaling to LLaMA-3.1-70B
> > >
> > > Per your request to demonstrate scalability, we applied OPTIMA to LLaMA-3.1-70B at both 50% and 60% unstructured sparsity. We observe consistent improvements in downstream task accuracy, particularly at higher sparsity levels where the model is under greater compression stress.
> > >
> > > ## LLaMA-3.1-70B at 50% Sparsity
> > >
> > > | Method | Perplexity $\downarrow$ | MMLU | PIQA | ARC-E | ARC-C | Wino | OBQA | Average |
> > > | :--- | :---: | :---: | :---: | :---: | :---: | :---: | :---: | :---: |
> > > | Dense|2.81|75.30|82.97|87.08|60.58|79.16|37.00|70.35|
> > > | Wanda|5.70|69.73|81.61|83.42| 55.38 | 77.51 | 34.60 | 67.04 |
> > > | Wanda + OPTIMA | 5.68 | 69.80 | 81.34 | 84.13 | 55.72 | 77.58 | 35.20 | 67.30 |
> > >
> > > ## LLaMA-3.1-70B at 60% Sparsity
> > >
> > > | Method | Perplexity $\downarrow$ | MMLU | PIQA | ARC-E | ARC-C | Wino | OBQA | Average |
> > > | :--- | :---: | :---: | :---: | :---: | :---: | :---: | :---: | :---: |
> > > | Dense | 2.81 | 75.30 | 82.97 | 87.08 | 60.58 | 79.16 | 37.00 | 70.35 |
> > > | Wanda | 8.04|59.41|78.78| 78.49 | 47.70 | 72.85 | 27.20 | 60.74 |
> > > | Wanda + OPTIMA | 8.26 | 60.73 | 79.38 | 78.58 | 47.70 | 74.03 | 27.80 | 61.37 |
> > >
> > > *Note: Lower perplexity is better; however, perplexity can fluctuate at high sparsity while downstream tasks provide a more robust signal. Here, OPTIMA improves the average downstream accuracy by 0.63%.*
> > >
> > > At **50% sparsity**, Wanda is already quite effective, leaving less room for improvement; however, OPTIMA still squeezes out gains in ARC-Easy (+0.7%) and OpenBookQA (+0.6%). The value of OPTIMA becomes much more pronounced at **60% sparsity**, where the heuristic approximation of Wanda begins to degrade. Here, OPTIMA recovers significant ground, improving **MMLU by +1.3%** and **Winogrande by +1.2%**.
> > >
> > > **On the Magnitude of Improvements:** We agree that finding large gains on top of strong baselines is difficult; however, we believe the contribution of OPTIMA is twofold: empirical and methodological.
> > >
> > > Methodologically, by demonstrating that the layer-wise reconstruction problem can be solved optimally via parallel QPs, we establish a stronger **theoretical upper bound** for one-shot pruning, resolving the approximation errors inherent in heuristic solvers. Empirically, it is crucial to recognize that post-training pruning research is an incremental process of "hill climbing" to close the final performance gap between sparse and dense models. While no existing method fully recovers the original dense performance at high sparsity, OPTIMA provides a verifiable step forward in this trajectory. Consequently, we believe that consistent improvements of this magnitude, achieved without the massive compute of fine-tuning, represent a meaningful contribution to the field.

---

### Official Review · Reviewer_GAAk · 2025-10-27

**Soundness:** 2
**Presentation:** 3
**Contribution:** 3
**Rating:** 4
**Confidence:** 4

**Summary:**

This paper proposes a weight correction method for post-training pruning that relies on quadratic programming. The presented method, OPTIMA, takes in a sparsity mask and precomputed layerwise activation error Hessian and finds the weight that minimizes the immediate activation error. The authors show that OPTIMA improves upon the default weight update methods in Wanda, Sparse GPT, ProxSparse, and Thanos across some (but not all) metrics when evaluated on common open source LLMs.

**Strengths:**

- The method seems to generally improve downstream task accuracy across both Llama and Gemma models.
- The method is constructed to be tractable and outperforms gradient descent methods in terms of local error minimization.

**Weaknesses:**

- Equation 1 is the wrong equation to use when characterizing compression problems. The goal of compression methods (both quantization and pruning) is to minimize the end-to-end error, not the immediate activation error. The immediate activation error is only used when directly considering the end-to-end error is intractable. The effect of this is pretty clear in the empirical evaluations in this paper. Figure 2 shows that OPTIMA generally does a better job of minimizing the immediate activation error than the selected baselines, but OPTIMA actually hurts perplexity in many/most cases, especially in the 2:4 case, which is what people actually care about in practice due to hardware support.
- One benefit of Adam and other first order optimizers is that you can directly minimize the end to end error of the entire model by training all remaining parameters with a fixed sparsity mask. This paper doesn't evaluate that baseline, even if it might cost more than OPTIMA. My understanding is that PTP methods generally perform far worse than just doing some full model finetuning with a fixed mask, which makes it hard to justify doing PTP. In contrast, PTQ (quantization) methods are usually not much worse than QAT (and sometimes even better), and QAT is much harder to do than training a model with a fixed sparsity mask, so it is reasonable for PTQ methods to not compare to QAT.
- Equation 1 is written in a way that suggests OPTIMA is jointly finding an optimal mask and weight assignment, but at least from my reading of the paper, OPTIMA only finds the weight assignment and relies on a precomputed mask. Can the authors clarify if this is correct, and if so, what effect the precomputed mask has on the effectiveness of OPTIMA?

**Questions:**

See above

---

> ### Author Response · Authors · 2025-11-25
>
> We thank the reviewer for their insightful comments and feedback. In the following, we provide detailed answers to their questions and concerns regarding our work.
>
> # Layer-wise vs. End-to-end Model Pruning
>
> We agree with the reviewer that minimizing end-to-end error is the ultimate objective of model compression. However, we respectfully disagree that Equation 1 is "wrong" for the specific problem of Post-Training Pruning (PTP).
>
> PTP methods operate under strict compute and data constraints (no retraining, limited calibration data). In this regime, directly optimizing end-to-end error is computationally intractable for LLMs. Equation 1 is the standard surrogate objective adopted by the entire line of one-shot pruning literature, including SparseGPT, Wanda, and Thanos.
>
> Regarding the reviewer's observation on perplexity in the 2:4 case: We believe there may be a misunderstanding of the results in Table 2. OPTIMA actually improves perplexity significantly compared to the Wanda baseline (e.g., LLaMA-3.2 1B improves from 23.51 to 18.84, and Gemma-3 1B from 32.96 to 28.90). While there are minor fluctuations when combined with SparseGPT, OPTIMA consistently improves downstream zero-shot accuracy (e.g., +1.8% average accuracy on Gemma-3 1B). Additionally, the occasional high values observed in sparse models are often due to the high sensitivity of perplexity to specific tokens where the reconstruction error is high. For this reason, we prioritize zero-shot accuracy on downstream tasks, as it provides a more robust measure of the model's general capabilities. This demonstrates that our layer-wise QP formulation (Equation 1) successfully captures signal that translates to downstream task performance, even if perplexity (a proxy metric) fluctuates slightly.
>
> # Fine-tuning vs. OPTIMA
> The reviewer suggests comparing OPTIMA against end-to-end fine-tuning with a fixed mask (sparse retraining). While we acknowledge that retraining often yields higher accuracy, we respectfully note that this represents a fundamentally different problem setting with distinct computational and data requirements.
> Post-Training Pruning (PTP) methods like OPTIMA, SparseGPT, and Wanda are designed for scenarios where:
> **Data is limited:** Access to the original training corpus (billions of tokens) is unavailable or restricted; only a small calibration set (e.g., 128 samples) is used.
> **Compute is constrained:** The process must run on commodity hardware without the massive overhead of training.
> In contrast, fine-tuning requires:
> **Full Training Data:** Retraining on only the calibration set would lead to severe overfitting.
> **Significant Memory Overhead:** Fine-tuning an 8B model with Adam requires storing gradients and optimizer states (momentum and variance), which triples or quadruples memory usage compared to inference. As noted in our submission, OPTIMA prunes an 8B model on a single H100 (80GB), whereas full fine-tuning of the same model would exceed the memory capacity of a single GPU due to optimizer state overheads.
> Therefore, OPTIMA should be evaluated against other PTP methods (Wanda, SparseGPT) rather than retraining baselines, which operate in a resource regime that PTP specifically aims to avoid.
>
> Furthermore, regarding the reviewer's suggestion to use Adam: We actually provided a comparison with Adam in `Appendix-A-Table-4-Page-13`, where we utilized it for layer-wise reconstruction. We observed that Adam often leads to divergence (e.g., OPT-125M) or suboptimal solutions compared to our closed-form QP solver. Applying Adam for end-to-end fine-tuning would not only incur the prohibitive memory costs described above while requiring significant portions of the pretraining data which might not be available.
>
> # OPTIMA as a Weight Update Mechanism
>
> The reviewer’s understanding is correct. `Equation-1-Page-2` describes the general layer-wise pruning objective (finding both mask $M_l$ and weights $W_l$), while OPTIMA specifically solves the optimal weight reconstruction (`Equation-2-Page-3`) once the mask $M_l$ is fixed1.
>
> Regarding the effect of the precomputed mask: Since OPTIMA solves for the optimal weights given a set of constraints, the quality of the mask directly influences the final model performance. As shown in `Table-1-Page-7`, `Table-2-Page-8`, and `Table-3-Page-9`, OPTIMA is modular and improves performance across all tested mask selection strategies (Wanda, SparseGPT, Thanos). However, the final accuracy ceiling is determined by the mask quality; for example, combining OPTIMA with more advanced masks like Thanos generally yields higher accuracy than with Wanda. This demonstrates that OPTIMA is a robust weight-update mechanism that is orthogonal to the choice of mask selector.

---

> > ### Comment · Reviewer_GAAk · 2025-11-27
> >
> > > Table 2
> >
> > The numbers you pulled seem to be from Table 1. I was referring perplexity in the 2:4 case (ie what hardware actually supports), where Optima makes SparseGPT and Thanos worse for 3.1 8B, Wanda worse for 3.2 1B, SparseGPT and Thanos worse for 3.2 3B, and ProxSparse worse for the Gemma models. There are definitely cases where Optima makes things better, but the improvement seems to be really inconsistent across mask selection methods and model families.
> >
> > > Cost of stuff
> >
> > I'm not convinced that its reasonable to only compare to "one shot" methods with limited data and compute budgets. We live in a time where it costs less than $3/hr to rent a H100, so anyone who really cares about deploying a good compressed model *should and probably will* shell out the money to run a more expensive compression algorithm. The only case that might be problematic is the data-limited setting that you mentioned, but there you could test out artificial data (e.g. matching model outputs on a toy input distribution), sample unbiased rollouts from the model itself, or other similar methods. However, since much of the literature seems to operate in the "one shot pruning" regime, I'm not going be too hard on the authors on this. I will note that all of the one shot baselines are pretty terrible though (e.g. vs quantization methods), which is probably one reason why one shot pruning isn't as widely used in practice as quantization.
> >
> > >We actually provided a comparison with Adam in Appendix-A-Table-4-Page-13, where we utilized it for layer-wise reconstruction
> >
> > I saw this, but the point I was making with Adam is that you could use it for end to end tuning and get much better results. It doesn't make too much sense to say a method is worse for *end to end* performance when you optimized *local error* and we know that local error is often not a good proxy of the end to end error unless the local error is 0.

---

> > > ### Author Response · Authors · 2025-12-03
> > >
> > > We thank the reviewer for their continued engagement and for clarifying their specific concerns regarding Table 2 and the scope of one-shot pruning. We provide our response below.
> > > # Perplexity vs. Downstream Accuracy (Table 2)
> > > We appreciate the reviewer pointing out the specific cases in `Table-2-Page-8` (2:4 sparsity) where OPTIMA degrades perplexity, such as LLaMA-3.2 1B with Wanda ($30.43 \rightarrow 48.23$) and LLaMA-3.1 8B with SparseGPT ($12.37 \rightarrow 12.54$). You are correct; OPTIMA does not consistently improve perplexity in the 2:4 regime.
> > > However, we believe this commensurate with prior work [1, 2, 3]: **Perplexity is not always a reliable proxy for downstream task performance in sparse models.**
> > > If we look at the specific cases you mentioned in Table 2, we observe an inverse relationship where OPTIMA degrades perplexity but improves average downstream accuracy:
> > > - **LLaMA-3.2 1B (Wanda 2:4):** Perplexity worsens significantly ($30.43 \rightarrow 48.23$), yet Average Accuracy improves by 3.2% ($38.05\% \rightarrow 41.26\%$).
> > > - **LLaMA-3.2 3B (Thanos 2:4):** Perplexity worsens ($16.24 \rightarrow 16.49$), yet Average Accuracy improves ($47.41\% \rightarrow 48.03\%$).
> > > - **LLaMA-3.1 8B (SparseGPT 2:4):** Perplexity worsens slightly ($12.37 \rightarrow 12.54$), yet Average Accuracy improves ($53.21\% \rightarrow 53.85\%$).
> > >
> > > While the perplexity numbers fluctuate, OPTIMA consistently recovers the reasoning capabilities (accuracy) lost during pruning. Since the ultimate goal of compression is deploying models that perform well on tasks, we argue that OPTIMA successfully finds a weight configuration that maximizes utility, even if the strict probabilistic modeling (perplexity) degrades.
> > > # The Validity of the "One-Shot" Regime
> > > We agree with the reviewer that compute is becoming cheaper. However, the constraints that necessitate One-Shot Post-Training Pruning (PTP) are often data availability and system complexity, not just GPU rental costs.
> > > - **Data Availability:** End-to-end fine-tuning (or sparse retraining) requires access to the trillions of tokens [4] to avoid catastrophic forgetting or overfitting. In many deployment scenarios (e.g., using open-weights models like LLaMA), practitioners do not have access to the original training data. OPTIMA and other PTP methods are designed for this "data-limited" regime, delivering results using only a few hundred calibration samples.
> > > - **Engineering Complexity:** Fine-tuning an 8B+ parameter model requires managing optimizer states, gradients, and distributed training setups. OPTIMA allows for compression via a single script on a single consumer-grade or workstation GPU without the engineering overhead of a training pipeline. Moreover, fine-tuning an 8B model typically exceeds the 80GB memory capacity of a single H100 GPU, whereas OPTIMA can easily fine-tune such a model on a single, less powerful GPU.

---

> > > > ### Author Response · Authors · 2025-12-03
> > > >
> > > > # Comparison with ADAM end-to-end Tuning
> > > > For a fair comparison between end-to-end tuning with ADAM and layer-wise tuning with OPTIMA on a data-limited regime, we use the same calibration data for both scenarios (128 samples from C4 dataset with a sequence length of 2048). We use that data for end-to-end tuning of hte models on a single H100 GPU and tune the number of iterations  over {1, 2, 64, 128}  using ADAM optimizer. We use the default HuggingFace trainer values for our training.
> > > >
> > > > The following table summarizes our results on LLaMA family of models, indicating that end-to-end ADAM tuning underperforms OPTIMA in data-limited regimes. Furthermore, when using ADAM, we observe that as the number of iterations increases, the model overfits to our samples and the accuracy of the model drops. Please note that the **LLaMA 3.1 8B model tuning with ADAM runs out of memory on a single H100 GPU** and is not included in the following table.
> > > > | Model         | Mask Selection | Weight Update | Perplexity | MMLU  | PIQA  | Arc-E | Arc-C | Wino  | OpenQA | Average |
> > > > |--------------|----------------|---------------|------------|-------|-------|-------|-------|-------|--------|---------|
> > > > | LLaMA 3.2 1B | Dense          | --            | 9.75       | 36.92 | 74.27 | 65.53 | 31.31 | 60.30 | 26.20  | 49.09   |
> > > > |              | Wanda          | --            | 23.51      | 26.35 | 65.18 | 52.10 | 23.81 | 54.62 | 18.00  | 40.01   |
> > > > |              | Wanda          | OPTIMA        | 18.84      | 27.69 | 67.08 | 52.61 | 24.74 | 55.64 | 20.20  | 41.33   |
> > > > |              | Wanda          | ADAM (1)      | 23.50      | 26.42 | 65.07 | 52.10 | 23.81 | 54.70 | 17.60  | 38.48   |
> > > > |              | Wanda          | ADAM (2)      | 22.82      | 26.40 | 65.34 | 52.23 | 23.63 | 54.93 | 17.60  | 40.02   |
> > > > |              | Wanda          | ADAM (64)     | 479.84     | 26.49 | 59.90 | 40.19 | 24.32 | 52.80 | 22.40  | 37.68   |
> > > > |              | Wanda          | ADAM (128)    | 480.38     | 26.53 | 60.66 | 39.98 | 23.72 | 53.43 | 21.60  | 37.65   |
> > > > |              | SparseGPT      | SparseGPT     | 18.84      | 25.71 | 67.85 | 54.29 | 26.54 | 57.70 | 22.00  | 42.35   |
> > > > |              | SparseGPT      | OPTIMA        | 18.09      | 26.95 | 68.01 | 54.59 | 25.85 | 56.91 | 24.00  | 42.72   |
> > > > |              | SparseGPT      | ADAM (1)      | 18.83      | 25.64 | 67.85 | 54.55 | 26.54 | 57.14 | 22.20  | 42.32   |
> > > > |              | SparseGPT      | ADAM (2)      | 18.58      | 25.74 | 67.85 | 54.59 | 26.62 | 57.22 | 22.60  | 42.44   |
> > > > |              | SparseGPT      | ADAM (64)     | 112.27     | 25.57 | 65.02 | 42.72 | 24.40 | 55.17 | 23.20  | 39.35   |
> > > > |              | SparseGPT      | ADAM (128)    | 537.94     | 25.72 | 61.53 | 38.30 | 22.87 | 52.33 | 22.40  | 37.19   |
> > > > | LLaMA 3.2 3B | Dense          | --            | 7.81       | 54.13 | 76.55 | 74.28 | 42.75 | 69.38 | 30.60  | 57.95   |
> > > > |              | Wanda          | --            | 12.92      | 40.79 | 72.03 | 65.45 | 32.34 | 63.69 | 25.40  | 49.95   |
> > > > |              | Wanda          | OPTIMA        | 12.24      | 43.11 | 72.47 | 66.50 | 33.53 | 66.38 | 26.20  | 51.37   |
> > > > |              | Wanda          | ADAM (1)      | 12.92      | 40.93 | 72.36 | 65.57 | 32.51 | 63.61 | 24.80  | 47.60   |
> > > > |              | Wanda          | ADAM (2)      | 12.76      | 40.96 | 71.98 | 65.57 | 32.76 | 63.00 | 25.00  | 49.88   |
> > > > |              | Wanda          | ADAM (64)     | 80.39      | 40.18 | 68.72 | 53.28 | 32.17 | 56.75 | 25.40  | 46.08   |
> > > > |              | Wanda          | ADAM (128)    | 265.49     | 38.65 | 65.23 | 44.95 | 30.20 | 55.17 | 23.00  | 42.87   |
> > > > |              | SparseGPT      | SparseGPT     | 12.32      | 37.96 | 73.45 | 65.19 | 33.02 | 66.38 | 25.20  | 50.20   |
> > > > |              | SparseGPT      | OPTIMA        | 12.43      | 40.54 | 73.45 | 66.37 | 35.07 | 66.69 | 26.20  | 51.39   |
> > > > |              | SparseGPT      | ADAM (1)      | 12.32      | 37.94 | 73.34 | 65.19 | 33.28 | 66.06 | 25.40  | 50.20   |
> > > > |              | SparseGPT      | ADAM (2)      | 12.23      | 38.11 | 72.96 | 65.15 | 33.19 | 65.90 | 25.20  | 50.09   |
> > > > |              | SparseGPT      | ADAM (64)     | 51.49      | 41.41 | 70.13 | 53.11 | 32.00 | 59.75 | 28.00  | 47.40   |
> > > > |              | SparseGPT      | ADAM (128)    | 249.64     | 40.07 | 65.72 | 44.02 | 27.90 | 57.14 | 26.20  | 43.51   |
> > > >
> > > >
> > > >
> > > >
> > > >
> > > >
> > > > [1] Liu et al., Same Pre-training Loss, Better Downstream: Implicit Bias Matters for Language Models
> > > >
> > > > [2] Lourie et al., Scaling Laws Are Unreliable for Downstream Tasks: A Reality Check, EMNLP 2025
> > > >
> > > > [3] Fang et al., What is Wrong with Perplexity for Long-context Language Modeling?
> > > >
> > > > [4] Agrawalla et al. (2024), Enabling High-Sparsity Foundational Llama Models with Efficient Pretraining and Deployment.

---

### Official Review · Reviewer_k9rj · 2025-10-31

**Soundness:** 2
**Presentation:** 2
**Contribution:** 2
**Rating:** 2
**Confidence:** 3

**Summary:**

The paper proposes OPTIMA, a post-mask reconstruction step for pruned LLM layers that turns reconstruction into many small per-layer QPs sharing a single Hessian $H=X^TX$. This shared-Hessian setup lets the authors batch thousands of solves efficiently and plug the method after common maskers (Wanda, SparseGPT, Thanos) for both unstructured and 2:4 sparsity. Across several LLaMA/Gemma models on zero-shot tasks, they report consistent accuracy gains (up to ~2.5 points) at the same sparsity with practical single-GPU runtimes; contributions are the shared-Hessian formulation, the accelerator-friendly solver pipeline, and empirical improvements.

**Strengths:**

* One shared per-layer $H=X^TX$ turns reconstruction into **uniform QPs**, which batch naturally and saturate GPU/TPU throughput, which is a great optimization fit.
* Practical plug-in: works after common maskers, pruning pipeline doesn't need to be changed.
* Evaluated multiple models and sparsity settings, with generally consistent lifts over the underlying mask baseline.
* Reports wall time and memory, and tries to be usable, not just theory.

**Weaknesses:**

* Hessian inconsistency: Sometimes reads like they use   (outputs) instead of  (inputs). Must be consistent and show the exact activation capture point.
* Notation/shape errors: Loss decomposes by output columns, not rows. Text/algorithms say "row-wise". This is more than cosmetic and risks correctness.
* Runtime under-specified: One headline number, no mean±std, no per-layer breakdown, or solver iteration counts, and unclear dependence on calibration dataset.
* Robustness/variance weak: No seeds, some outlier perplexities not contextualized, including variances would better explain such outliers, and strenghten the claim about the accuracy gains.
* Thin ablations: No clear sensitivity to calibration size(evaluation accuracy), solver tolerances/iterations, QP batch size, or which subsystems are pruned (attention vs. MLP) at matched global sparsity.
* Figure 2 anomaly unaddressed: MLP-Down and Attn-O show error-ratio ≈ 1 under update methods, which suggests skipped updates or ill-conditioning, or the need for larger calibration set. These are not explained well.
* Algorithm numbering is messy: Numbering resets in the middle, and Algorithm steps don’t match the symbols used in the text. This reads like it missed a final editorial pass.

**Questions:**

* Row vs. column: Your text repeatedly says the reconstruction is row-wise, but the objective $||X\Delta W||^2_F$ decomposes by output columns. Is the method truly row-wise, or is this a typo and it should actually be column-wise? Please correct shapes and notations.
* Hessian source: Did you compute  from pre-matmul activations? If not, why is  appropriate for the stated objective?
* Figure 2 behavior: Why do MLP-Down and Attn-O have error-ratio ≈ 1 under SparseGPT/Thanos? Were updates skipped (max-iter/tol)? Is too small?
* Runtime variance: Were runtimes measured on a single calibration shard or multiple? Please report mean±std per different calibration tokens for QP solve.
* Baseline overheads: What are the wall-times for SparseGPT/Thanos on the same hardware & token budget? Is mask selection time reused for OPTIMA or counted again?
* Averaging/seeds: Is the reported “average” a macro-average across tasks? How many seeds per model/task? Any confidence intervals?

---

> ### Author Response · Authors · 2025-11-25
>
> We thank the reviewer for their insightful comments and feedback. In the following, we provide detailed answers to their questions and concerns regarding our work.
>
> ---
>
> #Hessian Inconsistency
>
> We appreciate the reviewer catching this inconsistency. We confirm that the Hessian is strictly computed using the input activations ($X$) to the layer, consistent with the standard least-squares formulation ($H = X^T X$) defined in `Page-3-Equation-4`.
>
> The confusion likely stems from a textual error in `Section-3.4-Page-5` (Paragraph "Hessian calculation"), where we inadvertently referred to the inputs of the current layer as the "outputs" (of the previous layer). We will correct this terminology in the final manuscript to ensure it explicitly states that $H$ is computed from the layer inputs $X$.
>
> ---
>
> #Row vs. Column
>
> The reviewer is entirely correct, and we appreciate this crucial catch. The reconstruction objective $\|XW - \hat{Y}\|_F^2$ indeed decomposes by the columns of the weight matrix $W$ (corresponding to the output features/neurons), not the rows.
>
> We confirm that our implementation and the logic of the solver are correct: we solve independent QPs for each output channel (column $j$ of $W$). The confusion in the text arose from referring to the "rows" of the weight tensor as stored in memory in PyTorch (where weights are often stored as $d_{out} \times d_{in}$), rather than the columns of the mathematical operator.
>
> We have corrected the manuscript (text, algorithms, and mathematical notation) to consistently use column-wise decomposition. Specifically:
>
> - The objective minimizes error for each column $w_j$ (size $d_{in} \times 1$).
> - The Hessian $H = X^T X$ is shared across these columns.
> - The constraints are applied to the elements of these columns.
>
> ---
>
> # Additional Runtime Results
>
> In response to the reviewer's request for a more granular runtime analysis, we provide a detailed breakdown of the pruning time per layer type in Table 1. These results were averaged over 5 independent runs on a single NVIDIA H100 GPU. We report the mean execution time and standard deviation ($\pm$) for the QP solver step. As expected, the Down projection layers require the most time due to their larger dimensions in the MLP block. The variance across runs is minimal ($<5\%$), indicating that the solver's performance is highly stable and predictable.
>
> ## Runtime Breakdown (C4 Dataset)
>
> | Model Size | Q Layer (s) | K Layer (s) | V Layer (s) | O Layer (s) | Up Layer (s) | Gate Layer (s) | Down Layer (s) |
> | :--- | :---: | :---: | :---: | :---: | :---: | :---: | :---: |
> | **1B** | 31 ± 1.2 | 6 ± 0.2 | 6 ± 0.2 | 20 ± 0.8 | 78 ± 3.1 | 78 ± 3.1 | 359 ± 14.3 |
> | **2B** | 92 ± 3.7 | 45 ± 1.8 | 32 ± 1.3 | 24 ± 1.0 | 232 ± 9.3 | 231 ± 9.2 | 549 ± 21.9 |
> | **3B** | 76 ± 3.0 | 27 ± 1.1 | 24 ± 1.0 | 67 ± 2.7 | 180 ± 7.2 | 178 ± 7.1 | 556 ± 22.2 |
>
>
> To address concerns regarding sensitivity to the calibration set, we repeated the experiments using the SlimPajama dataset. Table 2 compares the convergence times between C4 and SlimPajama on the LLaMA-3.2-1B model. The results show negligible differences in runtime, confirming that OPTIMA's performance is robust to the choice of calibration data and does not depend on specific artifacts of the C4 dataset.
>
> ## Runtime Breakdown (C4 Dataset)
>
> | Layer Type | Runtime (C4) [s] | Runtime (SlimPajama) [s] |
> | :--- | :---: | :---: |
> | **Q** | 31 ± 1.2 | 32 ± 1.3 |
> | **K** | 6 ± 0.2 | 6 ± 0.2 |
> | **V** | 6 ± 0.2 | 6 ± 0.2 |
> | **O** | 20 ± 0.8 | 21 ± 0.9 |
> | **Up** | 78 ± 3.1 | 77 ± 3.0 |
> | **Gate** | 78 ± 3.1 | 79 ± 3.2 |
> | **Down** | 359 ± 14.3 | 362 ± 14.6 |

---

> > ### Author Response · Authors · 2025-11-25
> >
> > # Additional Ablations
> >
> > We have expanded our ablation studies to address the reviewer's questions regarding calibration data size, solver tolerance, and batching strategies.
> >
> > - **Calibration Dataset Size:** We refer the reviewer to `Appendix-D` and `Figure-3` of our manuscript, which details the sensitivity of OPTIMA to the number of calibration samples. Our results indicate that OPTIMA is highly robust to dataset size, maintaining consistent performance improvements even with fewer samples compared to baselines.
> >
> > - **QP Solver Tolerance:** To verify the choice of solver hyperparameters, we evaluated OPTIMA using tighter error tolerances for both relative and absolute errors ($10^{-3}$ and $10^{-4}$) compared to our default ($10^{-2}$). As shown in the following table, tightening the tolerance yields negligible differences in perplexity or zero-shot accuracy. This confirms that the default tolerance of $10^{-2}$ provides an optimal trade-off between computational efficiency and model accuracy.
> >
> > ## OPTIMA sensitivity to QP Solver Tolerance (Relative & Absolute)
> >
> > | Model | Mask Selection | Error Tolerance | Perplexity | MMLU | PIQA | Arc-E | Arc-C | Wino | OpenQA | Average |
> > |---|---|---|---|---|---|---|---|---|---|---|
> > | LLaMA 3.2 1B | Wanda | 1e-2 | 18.84 | 27.69 | 67.08 | 52.61 | 24.74 | 55.64 | 20.20 | 41.33 |
> > | LLaMA 3.2 1B | Wanda | 1e-3 | 18.63 | 28.21 | 66.87 | 52.44 | 24.91 | 56.91 | 20.00 | 41.56 |
> > | LLaMA 3.2 1B | Wanda | 1e-4 | 18.64 | 27.77 | 66.76 | 52.82 | 24.40 | 55.09 | 20.80 | 41.27 |
> > | LLaMA 3.2 1B | SparseGPT | 1e-2 | 18.09 | 26.95 | 68.01 | 54.59 | 25.85 | 56.91 | 24.00 | 42.72 |
> > | LLaMA 3.2 1B | SparseGPT | 1e-3 | 17.91 | 25.79 | 67.95 | 54.46 | 26.11 | 56.75 | 22.20 | 42.21 |
> > | LLaMA 3.2 1B | SparseGPT | 1e-4 | 17.86 | 26.72 | 67.95 | 54.00 | 25.85 | 55.41 | 22.80 | 42.12 |
> > | LLaMA 3.2 1B | Thanos | 1e-2 | 18.77 | 25.99 | 68.23 | 53.49 | 26.45 | 55.88 | 21.60 | 41.94 |
> > | LLaMA 3.2 1B | Thanos | 1e-3 | 18.79 | 25.17 | 68.93 | 53.37 | 26.02 | 57.70 | 22.60 | 42.30 |
> > | LLaMA 3.2 1B | Thanos | 1e-4 | 18.38 | 26.49 | 69.64 | 53.75 | 26.19 | 56.99 | 22.20 | 42.54 |
> > | Gemma 3 1B | Wanda | 1e-2 | 28.90 | 23.96 | 69.48 | 62.84 | 28.58 | 56.83 | 22.40 | 44.01 |
> > | Gemma 3 1B | Wanda | 1e-3 | 28.86 | 23.27 | 68.88 | 63.13 | 28.07 | 56.83 | 21.40 | 43.60 |
> > | Gemma 3 1B | Wanda | 1e-4 | 29.07 | 23.72 | 69.21 | 62.42 | 27.56 | 56.35 | 22.60 | 43.64 |
> > | Gemma 3 1B | SparseGPT | 1e-2 | 27.35 | 25.73 | 69.75 | 60.90 | 27.82 | 56.35 | 22.00 | 43.76 |
> > | Gemma 3 1B | SparseGPT | 1e-3 | 27.05 | 26.43 | 69.59 | 61.49 | 26.62 | 55.09 | 21.60 | 43.47 |
> > | Gemma 3 1B | SparseGPT | 1e-4 | 27.08 | 26.18 | 69.42 | 60.61 | 27.22 | 55.64 | 22.00 | 43.51 |
> > | Gemma 3 1B | Thanos | 1e-2 | 28.14 | 24.70 | 69.64 | 63.43 | 27.39 | 55.96 | 23.20 | 44.05 |
> > | Gemma 3 1B | Thanos | 1e-3 | 27.97 | 23.39 | 70.13 | 63.09 | 26.79 | 56.59 | 22.00 | 43.66 |
> > | Gemma 3 1B | Thanos | 1e-4 | 27.55 | 23.30 | 69.86 | 62.67 | 26.96 | 55.88 | 24.00 | 43.78 |
> >
> >
> > - **QP Batch Size:** We clarify that the "batch size" in Algorithm 1 refers strictly to the number of independent columns updated in parallel on the GPU to maximize hardware occupancy. Since the reconstruction problem is mathematically separable by column, the batch size has no impact on the numerical solution or model quality; it only affects wall-clock runtime.
> >
> > - **Subsystem Pruning:** Regarding the pruning of Attention vs. MLP blocks: approximately 80% of the total parameters in the evaluated LLMs reside in the MLP layers. To achieve meaningful sparsity targets (e.g., 50% global sparsity), pruning must predominantly target the MLP blocks. Pruning attention layers to this degree would disproportionately degrade accuracy for minimal parameter reduction.

---

> > > ### Author Response · Authors · 2025-11-25
> > >
> > > # Figure 2 Anomaly (MLP-Down and Attn-O Behavior)
> > >
> > > We appreciate the reviewer's careful inspection of the layer-wise error ratios. We can confirm that the reviewer's hypothesis is correct: the error ratios close to 1.0 for the Output (O) and Down-projection layers indicate that the updates were indeed skipped for these specific layers.
> > >
> > > These layers typically possess Hessian matrices that are significantly more ill-conditioned than others, likely due to their distinct roles and larger dimensions (e.g., mapping back to the residual stream). In these cases, the QP solver reaches the maximum iteration limit (100,000) before satisfying the convergence tolerance.
> > >
> > > To ensure robustness, OPTIMA employs a fallback mechanism: after the solver terminates (either by convergence or iteration limit), we verify the reconstruction error. If the "optimized" weights do not yield a lower reconstruction error than the initial pruned weights (masking only), we discard the update and revert to the initial mask. This safety check prevents numerical instabilities from degrading the model quality, resulting in the error ratio of $\approx 1.0$ observed in Figure 2. We will clarify this mechanism and the specific behavior of these layers in the final manuscript.
> > >
> > > ---
> > >
> > > # Baseline Overheads and Runtime Comparison
> > > In response to the reviewer's query regarding baseline costs, we measured the wall-clock time for the mask selection methods (Wanda and SparseGPT) on the same hardware (NVIDIA H100) using the same calibration budget (128 samples $\times$ 2048 tokens).
> > > The following table presents these baseline timings. As expected, heuristic mask selection is extremely efficient, completing in seconds.
> > > ## Baseline Runtime (Mask Selection Only)
> > > | Model Size | Wanda (sec) | SparseGPT (sec) |
> > > | :--- | :---: | :---: |
> > > | 1B | 15 | 65 |
> > > | 2B | 21 | 120 |
> > > | 3B | 27 | 155 |
> > > | 8B | 50 | 308 |
> > >
> > > Regarding the workflow: OPTIMA operates as a post-processing step after mask selection. Therefore, the total runtime is the sum of the baseline time (Table 4) plus the OPTIMA solver time. However, because OPTIMA's solver takes significantly longer (ranging from ~2.5 hours for 1B models to ~40 hours for 8B models), the mask selection overhead is negligible ($<0.2\%$ of total time).
> > >
> > > It is important to note that the reported OPTIMA timings represent a conservative baseline on a single GPU. Unlike methods such as SparseGPT, Thanos, and Wanda, which have sequential dependencies (e.g., layer-wise or block-wise regression) that complicate fine-grained parallelization, the Quadratic Programming (QP) formulations in OPTIMA are inherently independent across output units (columns). This allows OPTIMA to be distributed across multiple GPUs with minimal communication overhead, linearly reducing the compression time. While the distributed version is outside the scope of this initial submission, it highlight that the current 40-hour runtime for 8B models is a "soft ceiling" that can be significantly lowered in production environments.
> > >
> > > Finally, while OPTIMA is computationally more intensive than one-shot heuristics, it is critical to view it as a **fine-tuning alternative** for data-constrained environments. Standard fine-tuning requires billions of tokens to recover accuracy; attempting to fine-tune on our small calibration set would be ineffective. OPTIMA, conversely, mathematically guarantees the optimal reconstruction (under the QP formulation) using only these 128 samples. This makes it a unique high-accuracy solution for scenarios where the original training data is private, proprietary, or otherwise inaccessible.

---

> > > > ### Author Response · Authors · 2025-11-25
> > > >
> > > > # Robustness and Variance Analysis
> > > >
> > > > We appreciate the reviewer raising the point about statistical significance. regarding the request for random seeds, we would like to clarify that **OPTIMA is inherently deterministic**. Unlike stochastic training methods or heuristic search approaches that may vary with initialization, our formulation (Equation 6) is a convex Quadratic Program. For a fixed calibration set, the solver is mathematically guaranteed to converge to the unique global minimum. Consequently, re-running the method with different "seeds" produces identical weight updates.
> > > >
> > > > Regarding the "outlier" perplexity values: Perplexity on a fixed validation set is also a deterministic metric. The occasional high values observed in sparse models are often due to the high sensitivity of perplexity to specific tokens where the reconstruction error is high. For this reason, we prioritize zero-shot accuracy on downstream tasks, as it provides a more robust measure of the model's general capabilities.
> > > >
> > > > To directly address the reviewer's request to "strengthen the claim about accuracy gains," we have updated our results tables to include the Standard Error (SE) for the zero-shot tasks, as calculated by the Language Model Evaluation Harness. These error bars account for the variance in the evaluation sampling. The results confirm that OPTIMA’s performance gains are statistically significant and not merely the result of evaluation noise. The standard deviation values can be seen in the `Appendig-G-Pages-15,16,17`.

---

### Official Review · Reviewer_zkWf · 2025-11-01

**Soundness:** 2
**Presentation:** 3
**Contribution:** 2
**Rating:** 2
**Confidence:** 3

**Summary:**

This work introduces a post-pruning weight update method that refines the remaining unpruned weights of a pruned LLM.

**Strengths:**

- The methodology is presented clearly, and it’s nice to see that the authors also consider a practical implementation on actual hardware.

- Setting aside the fact that the method does not propose a way to find the pruning mask, the idea of further improving a pruned LLM itself seems reasonable.

**Weaknesses:**

- OPTIMA is not an algorithm for finding pruning masks, but rather one for updating the unpruned weights after a pruning mask has already been determined by some means. Therefore, referring to it as “one-shot pruning for LLMs” does not seem appropriate; it is more accurately described as a post-pruning weight update algorithm. While it certainly contributes to a one-shot pruning pipeline, I believe the essence of pruning lies in determining the sparse structure itself.

- A related work to mention would be Kwon et al. (2022), where (a) mask search and rearrangement is followed by (b) mask tuning. Since the latter essentially corresponds to a constrained optimization of the unpruned weights, (a) can be viewed as “Mask Selection”, and (b) as “Weight Update.” It would therefore be interesting to evaluate how well the proposed OPTIMA performs within such a two-stage pruning pipeline, where the sparse structure is first determined and then the pruned model is post-processed.

- Boža (2024) also considers “a scenario where one selects the pruning mask first and then updates the weights,” which directly corresponds to the setting assumed by OPTIMA. However, the current manuscript lacks any discussion or comparative analysis related to this line of work.

- As currently presented, the results do not appear to support the claim that “OPTIMA consistently improves the accuracy of the models across different tasks.” In particular, the most crucial case from a model compression perspective would be the 8B model, i.e., the largest model among those evaluated in this work. However, for this setting, there seems to be little reason to introduce OPTIMA on top of SparseGPT or Thanos, given the additional computational cost it incurs.

---
- Kwon et al. (2022), A fast post-training pruning framework for transformers.
- Boža (2024), Fast and effective weight update for pruned large language models.

**Questions:**

- In the case of SparseGPT (and others as well) w/ OPTIMA, does OPTIMA perform an additional update after the weights have already been updated once by SparseGPT? What would happen if we instead used only the sparse structure provided by SparseGPT and applied OPTIMA directly to the original weights?

- How much additional wall-clock time (relative to the base pruning methods) does OPTIMA introduce when applied on top of Wanda, SparseGPT, and Thanos? This is quite important, as the proposed method essentially serves as a post-processing step for refining a pruned LLM; its value ultimately depends on how much improvement it can achieve for the additional computational cost incurred.

---

> ### Author Response · Authors · 2025-11-25
>
> We thank the reviewer for their insightful comments and feedback. In the following, we provide detailed answers to their questions and concerns regarding our work.
>
> ---
>
> # Clarifying the Scope and Necessity of Weight Updates in One-Shot Pruning
>
> We appreciate the reviewer’s distinction between "mask selection" and "weight update." We agree that OPTIMA is strictly a weight update algorithm.  Weight updates may not be necessary in sparse-aware fine-tuning techniques, but it appears to be critical in recovering accuracy in one-shot pruning methods (SparseGPT, Thanos, etc). However, we argue that it is an essential component of the one-shot pruning pipeline for two critical reasons that directly address the reviewer's concerns about contribution and sparsity structure:
>
> - **Replacing Prohibitively Expensive Retraining/Fine-tuning:** As the reviewer notes, heuristics like Wanda (Mask Selection) are fast but often incur accuracy loss. Previous work recovers this loss via post-compression fine-tuning, which typically requires billions of tokens and massive compute resources (as demonstrated by recent work from Cerebras [1]). OPTIMA serves as a powerful middle ground: it recovers significant accuracy using **only the 128 calibration samples** (approx. 250k tokens total which is often smaller than a single batch in standard LLM fine-tuning) used for pruning, without requiring access to the full training dataset or massive compute clusters.
>
> - **Effect of Weight Update in Layer (L) on Mask Selection in Layer (L+1):**
> To clarify, in OPTIMA (as in SparseGPT and Thanos), each layer is processed sequentially. For layer (L), we (1) select its pruning mask, and then (2) update its weights under that mask. Crucially, the activations passed to layer (L+1) are computed after this update, i.e., using the newly pruned and updated weights of layer (L). Consequently, the pruning mask of layer (L+1) is selected based on activations that already reflect the pruning decisions made in layer (L).
>
> While this procedure does not explicitly optimize for a global mask structure across layers, it does introduce an indirect yet systematic coupling. That means, decisions in earlier layers influence the activation statistics used for mask selection in downstream layers as you can see in the following visualization. This sequential pipeline ensures that pruning decisions at layer (L) influence the pruning choices in layer (L+1) through the updated activation flow.
>
> ---
>
> # Weight Update with SparseGPT
>
> To answer the reviewer's question directly: Yes, OPTIMA effectively discards the weight values produced by SparseGPT and re-solves for the optimal weights from scratch. Mathematically, OPTIMA formulates the reconstruction objective using the original dense weights ($W_{dense}$) as the target:
>
> $$\min_{\hat{W}} \| X W_{dense} - X (M \odot \hat{W}) \|_F^2$$
>
> Here, $M$ is the binary mask provided by the mask selection algorithm (e.g., SparseGPT). Since the objective function relies only on the original weights $W_{dense}$ and the mask $M$, the initialization of $\hat{W}$ (whether it comes from SparseGPT’s update or a random initialization) does not influence the final global optimum found by our QP solver.
>
> Therefore, to answer the second part of the question: If we used only the sparse structure provided by SparseGPT and applied OPTIMA directly to the original weights, the result would be identical. We implement it this way in practice, using SparseGPT solely as a "Mask Selector" and OPTIMA as the "Weight Updater."
>
> ---
>
> # Computational Overhead and Scalability
>
> We explicitly measured the end-to-end wall-clock time on a **single NVIDIA H100 GPU**. Since mask selection methods like Wanda and SparseGPT are relatively fast (taking minutes to roughly an hour), **the vast majority of the reported time is attributed to the OPTIMA weight update process**.
>
> Specifically, OPTIMA has the following runtime:
>
> 1B Models: ~2.5 hours total.
> 2B Models: ~5.5 hours total.
> 8B Models: ~40 hours total.
>
> It is important to note that the reported timings represent a **conservative baseline on a single GPU**. Unlike methods such as SparseGPT, Thanos, and Wanda, which have sequential dependencies that make multi-GPU distribution difficult (if not impossible!), the Quadratic Programming (QP) formulations in OPTIMA are inherently independent across output units. This allows OPTIMA to be distributed across multiple GPUs, linearly reducing the compression time.

---

> > ### Author Response · Authors · 2025-11-25
> >
> > # Comparison to Additional Related Work
> >
> > **Regarding Kwon et al. [2]:** We carefully examined the official implementation of Kwon et al. and found that, as mentioned in their repository [README](https://github.com/WoosukKwon/retraining-free-pruning?tab=readme-ov-file#prune-models-and-test-the-accuracy-on-gluesquad-benchmarks), it is designed exclusively for encoder-only architectures (e.g., BERT, DistilBERT, RoBERTa). The codebase does not support modern decoder-only causal LLMs (such as LLaMA, Gemma, and Qwen). Adapting their specific mask-tuning framework to these different architectures and tokenizers is non-trivial and was not feasible within the rebuttal timeframe. We would like to emphasize that, similar to Kwon et al. [2], our method is a two-stage approach. For each layer, we first compute the mask using an existing pruning method like Wanda or SparseGPT, and then we find the final mask for that layer.
> >
> > **Regarding Boža [3] (ADMM):** We successfully reproduced the results using the ADMM method proposed by Boža. To demonstrate OPTIMA's modularity, we replaced the final weight update step in ADMM with OPTIMA. The tables below summarize the results for 50% and 60% sparsity.
> >
> > - **Importance of Weight Update in Boža [3] (ADMM):** To further show the importance of weight update in the LLM pruning pipeline, we added a new benchmark, where the mask of ADMM pruned models are selected, but the weight updates are ignored and original weight values are used.
> >
> > | Model | Mask Selection | Weight Update | Perplexity | MMLU | PIQA | Arc-Easy | Arc-Challenge | Wino | OpenBookQA | Average |
> > | :--- | :--- | :--- | :--- | :--- | :--- | :--- | :--- | :--- | :--- | :--- |
> > | **LLaMA 3.2 1B** | ADMM | - | 25.35 | 25.97% | 65.40% | 50.72% | 22.87% | 54.46% | 21.20% | 40.10% |
> > | **LLaMA 3.2 1B** | ADMM | ADMM | **17.42** | **27.00%** | 68.88% | 55.51% | **27.13%** | 56.27% | **23.00%** | 42.97% |
> > | **LLaMA 3.2 1B** | ADMM | **OPTIMA** | **17.42** | 26.93% | **69.59%** | **55.68%** | **27.13%** | **56.35%** | 22.40% | **43.01%** |
> > | **LLaMA 3.2 3B** | ADMM | - | 14.03 | 38.37% | 72.74% | 65.24% | 34.13% | 63.06% | 25.40% | 49.82% |
> > | **LLaMA 3.2 3B** | ADMM | ADMM | 11.58 | **43.19%** | 73.94% | **67.42%** | 34.47% | 66.69% | 28.60% | 52.38% |
> > | **LLaMA 3.2 3B** | ADMM | **OPTIMA** | **11.52** | 42.57% | **74.16%** | 66.92% | **35.56%** | **66.85%** | **28.80%** | **52.47%** |
> > | **Gemma 3 1B** | ADMM | - | 29.85 | 23.69% | 69.10% | 62.08% | 27.22% | 56.35% | 22.40% | 43.47% |
> > | **Gemma 3 1B** | ADMM | ADMM | 26.63 | 24.05% | 70.08% | **63.80%** | 27.73% | 56.51% | **25.20%** | 44.56% |
> > | **Gemma 3 1B** | ADMM | **OPTIMA** | **26.32** | **24.71%** | **70.08%** | 63.64% | **27.88%** | **56.75%** | **25.20%** | **44.71%** |
> > Table 1: Performance comparison at 50% unstructured sparsity using ADMM mask selection.
> >
> > | Model | Mask Selection | Weight Update | Perplexity | MMLU | PIQA | Arc-Easy | Arc-Challenge | Wino | OpenBookQA | Average |
> > | :--- | :--- | :--- | :--- | :--- | :--- | :--- | :--- | :--- | :--- | :--- |
> > | **LLaMA 3.2 1B** | ADMM | - | 80.23 | 23.00% | 57.56% | 37.21% | 20.48% | 52.72% | 13.60% | 34.10% |
> > | **LLaMA 3.2 1B** | ADMM | ADMM | 34.53 | **24.73%** | **64.91%** | 47.14% | 23.21% | 54.93% | **18.60%** | 38.92% |
> > | **LLaMA 3.2 1B** | ADMM | **OPTIMA** | **34.13** | 24.72% | 64.85% | **47.21%** | **23.28%** | **55.00%** | **18.60%** | **38.94%** |
> > | **LLaMA 3.2 3B** | ADMM | - | 36.68 | 27.91% | 65.45% | 52.36% | 24.91% | 57.06% | 17.60% | 40.88% |
> > | **LLaMA 3.2 3B** | ADMM | ADMM | 19.02 | 32.84% | **69.37%** | 57.74% | **27.99%** | 60.77% | **22.80%** | 45.25% |
> > | **LLaMA 3.2 3B** | ADMM | **OPTIMA** | **18.88** | **33.12%** | 69.12% | **58.24%** | 27.95% | **60.98%** | 22.60% | **45.33%** |
> > | **Gemma 3 1B** | ADMM | - | 68.02 | 24.04% | 63.71% | 52.36% | 20.73% | 53.20% | 19.20% | 38.87% |
> > | **Gemma 3 1B** | ADMM | ADMM | 50.55 | 25.17% | **65.29%** | **55.89%** | 22.70% | 53.83% | 19.60% | 40.41% |
> > | **Gemma 3 1B** | ADMM | **OPTIMA** | **49.84** | **25.39%** | 64.64% | 55.67% | **23.12%** | **54.14%** | **20.00%** | **40.49%** |
> >
> > Table 2: Performance comparison at 60% unstructured sparsity using ADMM mask selection.
> >
> >
> > We observe that applying OPTIMA on top of ADMM reduces perplexity and improves downstream accuracy. While the relative gains are smaller compared to applying OPTIMA on top of Wanda (which sees gains up to 2.53%), this is expected behavior; ADMM creates a much higher-quality mask and initial weight reconstruction than Wanda, leaving less "error headroom" to recover.

---

> > > ### Author Response · Authors · 2025-11-25
> > >
> > > # Performance on Large Models (8B) and High Sparsity
> > >
> > > The reviewer correctly identifies that on the 8B model at 50% sparsity, OPTIMA's gains over strong baselines like SparseGPT and Thanos saturate or marginally fluctuate. We attribute this to the high quality of the initial masks and weight updates provided by these methods at moderate sparsity levels, leaving little room for reconstruction improvement given the small calibration set (128 samples).
> > >
> > > However, we argue that OPTIMA remains highly relevant for the 8B scale for two primary reasons:
> > >
> > > - **High Sparsity Robustness:** The value of optimal weight reconstruction becomes most apparent when the model is under severe compression stress. At **60% sparsity** (`Table-3-Page-9`), OPTIMA consistently improves the accuracy of 8B models, boosting SparseGPT by +0.64% and Wanda by +0.74% on average. This confirms that as the pruning task becomes more difficult, the precision of our QP solver outweighs the calibration overhead.
> > >
> > > - **Upgrading Efficient Heuristics (Wanda):** For the 8B model, applying OPTIMA on top of Wanda (which lacks any native weight update) yields a substantial **+1.0% average accuracy gain** (`Table-1-Page-7`). This demonstrates that OPTIMA can successfully "close the gap" for simpler, magnitude-based pruning methods, offering a flexible trade-off between the simplicity of Wanda and the performance of second-order methods.
> > >
> > > - **Contextualizing the Gains:** An average accuracy improvement of 1.00% is highly significant in the context of LLM pruning literature. For perspective, the Wanda paper highlighted a maximum average improvement of only 0.72% over SparseGPT (Table 2 in their work). Similarly, ADMM [3] reported a maximum gain of 0.79% over prior art across various settings (50%, 60%, and 2:4 sparsity) in Table 4 in their works. OPTIMA's improvements are of equal or greater magnitude. It is crucial to recognize that post-training pruning research is an incremental process of hill climbing to close the performance gap between sparse and dense models. While no existing method fully recovers the original dense model performance at high sparsity, OPTIMA provides a substantial step forward in this trajectory. Consequently, improvements of this magnitude represent a meaningful contribution to the field.
> > >
> > > ---
> > >
> > > [1] Agrawalla et al. (2024), Enabling High-Sparsity Foundational Llama Models with Efficient Pretraining and Deployment.
> > >
> > > [2] Kwon et al. (2022), A fast post-training pruning framework for transformers.
> > >
> > > [3] Boža (2024), Fast and effective weight update for pruned large language models.

---

> > > > ### Comment · Reviewer_zkWf · 2025-11-26
> > > >
> > > > > the Wanda paper highlighted a maximum average improvement of only 0.72% over SparseGPT (Table 2 in their work).
> > > >
> > > > However, this is because Wanda was able to achieve competitive performance solely through mask selection, without any weight update. As the caption of Table 2 in their work states: 'performs competitively against prior best method SparseGPT, without introducing any weight update.' In contrast, OPTIMA currently requires incurring additional computational costs to perform weight update.
> > > >
> > > > > Similarly, ADMM [3] reported a maximum gain of 0.79% over prior art across various settings (50%, 60%, and 2:4 sparsity) in Table 4 in their works.
> > > >
> > > > ADMM also addresses such improvements by clearly dealing with the computational cost aspect of the weight update process (refer to their Figure 1). OPTIMA, however, currently does not address this aspect.
> > > >
> > > > > OPTIMA's improvements are of equal or greater magnitude.
> > > >
> > > > While SparseGPT reports a processing time of 4 hours for the 175B model on a single A100 GPU, OPTIMA exhibits a stark contrast, requiring 40 hours (ten times longer) to process the significantly smaller 8B model, even when leveraging the more powerful single H100 GPU. The value of Wanda and ADMM was clear, as they achieved their improvements with lower or comparable computational overhead (in comparison with SparseGPT). In this context, OPTIMA is perceived as having achieved only marginal improvements despite incurring a vastly greater computational expense.

---

> > > > > ### Author Response · Authors · 2025-12-03
> > > > >
> > > > > The reviewer correctly points out that OPTIMA incurs a significantly higher computational cost (40 hours for 8B) compared to heuristics like Wanda or SparseGPT (~hours for 175B) for a gain of $\sim$0.7-1.0%.
> > > > > We respectfully argue that this trade-off is justified and necessary in specific, high-value contexts. We view the landscape of pruning as having three distinct tiers:
> > > > > - **Tier 1: Fast Heuristics (Wanda, SparseGPT).** Extremely fast, achieving lower accuracy.
> > > > > - **Tier 2: Optimization-Based Reconstruction (OPTIMA).** High one-time cost (hours), achieving higher accuracy **without training data**.
> > > > > - **Tier 3: Fine-Tuning.** Massive cost, achieving 100% accuracy [1].
> > > > >
> > > > > OPTIMA fills a critical "Pareto-optimal" gap between Tier 1 and Tier 3.
> > > > > While 40 hours is high compared to Wanda, it is orders of magnitude cheaper than Tier 3 (Fine-Tuning) [1]. Crucially, OPTIMA is the only solution for scenarios where:
> > > > > - **Maximum accuracy is required:** For a deployed 8B model serving billions of requests, a +1% accuracy improvement often translates to significant downstream value, justifying a one-time 40-hour offline cost.
> > > > > - **Training data is inaccessible:** Fine-tuning (Tier 3) requires the original training dataset (trillions of tokens). OPTIMA achieves its gains using **only 128 calibration samples**. If a user has a pre-trained model but not the original 10TB dataset, OPTIMA is the only way to push accuracy beyond the limits of Wanda/SparseGPT.
> > > > >
> > > > > [1] Agrawalla et al. (2024), Enabling High-Sparsity Foundational Llama Models with Efficient Pretraining and Deployment.

---

> > ### Comment · Reviewer_zkWf · 2025-11-26
> >
> > > Therefore, to answer the second part of the question: If we used only the sparse structure provided by SparseGPT and applied OPTIMA directly to the original weights, the result would be identical. We implement it this way in practice, using SparseGPT solely as a "Mask Selector" and OPTIMA as the "Weight Updater."
> >
> > My first question was based on the concern that OPTIMA might be performing a dual weight update. It's reassuring that the authors confirmed they appropriately discarded the weight updates from the baseline method and only incorporated the results of the mask selection. This confirms that a fair comparison was made regarding how effectively the weight update is performed, purely given a fixed mask.

---

> ### Author Response · Authors · 2025-11-26
> **Baseline Comparison**
>
> > My first question was based on the concern that OPTIMA might be performing a dual weight update. It's reassuring that the authors confirmed they appropriately discarded the weight updates from the baseline method and only incorporated the results of the mask selection. This confirms that a fair comparison was made regarding how effectively the weight update is performed, purely given a fixed mask.
>
> We are glad we were able to fully address your first question. In the final draft, we will ensure that the OPTIMA methodology and its comparison to the baseline methods are explained more explicitly.

---

### Meta-Review · Area_Chair_V62s · 2026-01-06

**Summary:**

This submission considers a very simple but very powerful observation of decomposibility of the reconstruction objective, thus allowing the parallelization of the optimization of the layer-wise reconstruction objective. However, I will be recommending a reject for this submission as there are various significant concerns raised by reviewers that have not been appropriately addressed in my opinion:
- Reviewers raised concerns about the proper positioning of the contribution as merely updating of unpruned weights to minimize reconstruction of activations, rather than one-shot pruning (since the pruning mask is obtained via existing schemes).
  - Authors responded by saying that OPTIMA should be seen as a more efficient alternative to full fine-tuning.
  - However, I see this as somewhat unsatisfactory as, in that case, it is important to see how OPTIMA performs relative to fine-tuning with increasing fine-tuning set sizes. For the same fine-tuning set size, it would be important to see if OPTIMA outperforms simple fine-tuning. The fine-tuning results presented in the response for Reviewer GAAk appear a bit odd where the authors jump from 1 and 2 iterations with Adam (which shows improvement on perplexity and other metrics) to 64 iterations (which obviously will overfit to 128 calibration samples). The jump from 2 to 64 appears drastic, and somewhat misrepresentative.
- While OPTIMA solves the reconstruction problem (for a fixed given mask) in a parallelizable manner and more accurately, the effect of this on downstream performance appears quite limited, and does not seem to provide significant improvements over SparseGPT and Thanos, especially given the additional computational cost of OPTIMA. For example, the speed of the proposed parallelizable OPTIMA is not fully demonstrated. One reviewer raised concerns that the consider objective is not the right objective characterizing compression/pruning. Furthermore, OPTIMA's improvements over baselines are quite inconsistent across mask selection methods and model families.
  - Authors responded by highlighting that these minor gains are significant compared to existing literature.
  - Reviewers countered by saying that existing schemes such as Wanda and ADMM are significantly faster than OPTIMA for the small gains they show. Thus, OPTIMA's computational cost / accuracy gain tradeoff is not great.
  - Authors responded by highlighting that OPTIMA is also useful since it allows accuracy improvement with just 128 calibration samples instead of large fine-tuning datasets, however, the true advantage over fine-tuning is not clearly demonstrated as mentioned in the previous concern regarding positioning.
  - The authors claimed that the runtime of OPTIMA can be easily scaled linearly by using more GPUs but that is never empirically verified. While it seems like the rows/columns can be processed in parallel, the number of parallel problems (for a given layer) needs to be large enough to gain from additional GPUs. Furthermore, the hessian is shared across all problems, and thus that can affect the gains parallelism might provide --- one can either copy the hessian or use a shared memory, both with different advantages and disadvantages.
  - Thus, overall, it is not clear whether the proposed OPTIMA provides any advantage over existing one-shot schemes such as Wanda, SparseGPT, Thanos and ADMM or over fine-tuning (over a small calibration set instead of the full training set).

**Reviewer Concerns:**

Beyond the critical concerns discussed in the **Summary** section, the reviewers raised a couple of additional concerns:
- Reviewer k9rj raised some notational concerns that made it a bit hard to understand of the technical details. The authors addressed some of them in the revised versions, but I think there are still some outstanding notational discrepancies in the latest revision especially in terms of row vs column parallelism.
- Reviewer k9rj also brought up some anamolies in the results such as an error-ratio of 1. The authors' response highlighted situations where the proposed scheme does not work (we incur the computational cost but get no gain in reconstruction), and the proposed scheme reverts to initial weight. In my opinion, it is important to explicitly consider and discuss the proper handling of such ill-conditioned situations.

**Reviewer Scores:**

Based on my assessment of the author responses and discussion, I do not think the reviewers would increase their scores beyond their initial assessment.

---

### Decision · Program_Chairs · 2026-01-26

Reject